# Influence Estimation in Statistical Models Using the Fisher Information Matrix

**Omri Y. Lev**                                                                 *omrilev@mit.edu*

*Department of Electrical Engineering and Computer Science*
Massachusetts Institute of Technology

**Ashia C. Wilson**                                                             *ashia07@mit.edu*
*Department of Electrical Engineering and Computer Science*
Massachusetts Institute of Technology

**Reviewed on OpenReview:** https://openreview.net/forum?id=1kYIBaXCG8

## Abstract

Quantifying how infinitesimal perturbations of training data affect a model is key to diagnosing and improving learning systems. This task was addressed via the notion of influence functions (Hampel, 1974; Koh and Liang, 2017; Koh et al., 2019; Bae et al., 2022). Following classical works, whenever the underlying problem can be cast as a weighted empirical risk minimization problem, many such influence estimators rely on the Fisher Information Matrix (FIM). Following these lines, we provide a new accuracy analysis that characterizes the asymptotic behavior of FIM-based influence estimators and compare these to Hessian-based influence estimators, while we further extend the theory to objectives with non-differentiable regularizers. The results obtained are broadly applicable and admit an efficient algorithm with favorable computational complexity. Simulations on realistic setups demonstrate its usefulness in terms of accuracy and computational efficiency in many learning settings.

## 1 Introduction

Understanding how a model's behavior changes with slight modifications to its training data is crucial for numerous machine-learning applications. These include detecting harmful patterns and constructing adversarial examples (Koh and Liang, 2017; Koh et al., 2019; Basu et al., 2020), conducting efficient cross-validation (CV) for model assessment and model selection (Beirami et al., 2017; Wilson et al., 2020), enabling data unlearning without full retraining (Sekhari et al., 2021; Wilson et al., 2020), and evaluating robustness to data-dropping (Broderick et al., 2020), among others. A common foundation for these tasks is the use of second-order approximations to capture the model's sensitivity to training data perturbations.

A classical technique for this task is the Newton step, which leverages a gradient preconditioned by the inverse Hessian matrix. However, this approach can be computationally prohibitive and numerically unstable, particularly in high-dimensional and non-convex scenarios (Kunstner et al., 2019; Bae et al., 2022). Several studies have explored influence approximations based on variants of the Fisher Information Matrix (FIM) (Singh and Alistarh, 2020; Sattigeri et al., 2022; Bae et al., 2022; Grosse et al., 2023; Choe et al., 2024). Yet, despite growing empirical adoption, there remains a lack of theoretical understanding to guide the selection and use of FIM variants across diverse applications. Many of these works rely exclusively on the empirical FIM, which is known to underperform in several settings. Moreover, prior theoretical analyses of influence functions have largely assumed smooth, differentiable regularization, most commonly $\ell_2$, which limits their applicability in practical settings. Indeed, modern machine learning models frequently incorporate non-differentiable regularizers (e.g., $\ell_1$/elastic net/group sparsity penalties, etc.), and recent work has shown that

---

Corresponding Author: omrilev@mit.edu

even certain neural networks can be framed as convex optimization problems with non-smooth regularization terms (Pilanci and Ergen, 2020; Zeger and Pilanci, 2025). Additionally, given recent efforts in multiple works regarding influence estimation with non-differentiable regularization (Wilson et al., 2020; Suriyakumar and Wilson, 2022), it remains unclear whether similar results can be obtained for influence functions that use the FIM instead of the Hessian in their computations.

In this work, we propose the Approximate Fisher Influence Function (AFIF), a practical and theoretically justified framework for estimating influence in statistical models. AFIF uses the Fisher information matrix derived from an exponential family structure, offering a computationally efficient alternative to Hessian-based methods. Unlike prior influence techniques, which often fail to handle general regularization and lack formal guarantees for FIM-based approximations, our approach is provably accurate in convex settings and supports a wide class of regularization types, including non-differentiable ones.

**Contributions:** We provide new accuracy guarantees for influence estimators that are based on the FIM under standard convexity and regularity conditions, while *not* requiring differentiable regularizers. This yields a principled extension of several influence frameworks (e.g., (Giordano et al., 2019b; Wilson et al., 2020; Sekhari et al., 2021; Suriyakumar and Wilson, 2022)), upgrading them to a computationally efficient FIM-based estimator that remains applicable with non-differentiable regularization. We demonstrate our newly developed theory via a set of experiments that correspond to different influence measurement settings.

**Notation:** Random variables are represented by sans-serif fonts $(\mathsf{x}, \mathsf{y}, \mathsf{z})$, and their realizations by regular italics $(x, y, z)$. The PDF of $\mathsf{z}$ is $P_{\mathsf{z}}(\cdot)$. Sets of values are indicated by capital calligraphic letters, such as $\mathscr{D} \triangleq \{z_1, z_2, \ldots, z_n\}$. Matrices are in bold capitals, with $\mathbb{I}_d$ as the $d \times d$ identity matrix. We use $f(x) = o(g(x))$ and $f(x) = O(g(x))$ when $f(x)/g(x) = 0$ and $f(x)/g(x) = c \neq 0$ in the limit $x \to \infty$. We denote the Lipschitz constant of a function $f$ by $\mathrm{Lip}(f) \triangleq \sup\{\|f(x) - f(y)\| \, / \, \|x - y\| : x \neq y \in \mathrm{supp}(f)\}$. The inner product between two vectors $\theta_1$ and $\theta_2$ is denoted by $\theta_1^\top \theta_2 \triangleq \langle \theta_1, \theta_2 \rangle$.

## 2   Problem Statement

Given a dataset $\mathscr{D} = (z_1, z_2, \ldots z_n)$ where each $z_i$ is comprised of a covariate $x_i$ and a label or response $y_i$, it is commonplace to use empirical risk minimization (ERM) to obtain a predictive model to deploy. In this work, we consider the problem of *weighted* ERM (wERM), i.e. given a loss function $\ell(\cdot)$, regularizer $\pi(\cdot)$, regularization parameter $\lambda \in [0, \infty)$ and weights $w^n \triangleq (w_1, \ldots, w_n)$, our goal is to find $\hat{\theta}(w^n)$, defined by

$$\hat{\theta}(w^n) \triangleq \operatorname*{argmin}_{\theta} L(\mathscr{D}, \theta, \lambda, w^n), \tag{1}$$

$$L(\mathscr{D}, \theta, \lambda, w^n) \triangleq \frac{1}{n} \sum_{i=1}^{n} w_i \ell(z_i, \theta) + \lambda \pi(\theta).$$

This formulation is equivalent to ERM when $w^n = (1, \ldots, 1) \triangleq \vec{1}$, whose solution is denoted by $\hat{\theta}(\vec{1})$ [1].

In many scenarios $\ell(z, \theta) = -\log(P(y|f(x; \theta)))$; that is, the loss can be interpreted as a negative log-likelihood under a probabilistic model induced by a parameterized function $f(x; \theta)$, often taken to be a neural network. Moreover, we study the case where $P(y|f(x; \theta))$ belongs to an *exponential-family* (Wainwright et al., 2008) whose natural parameters are $f(x; \theta)$ and whose natural statistics are denoted by $t(y)$, namely, $\log(P(y|f(x; \theta))) = f(x; \theta)^\top t(y) - \log(\sum_{\tilde{y} \in \mathcal{Y}} \exp(f(x; \theta)^\top t(\tilde{y}))) + \beta(y)$ for some function $\beta(y)$. This is satisfied by many common loss functions in machine learning (see popular examples for such losses in App. B).

*Remark* 1. Following (Banerjee et al., 2005), this class of losses corresponds to functions that can be captured by a Bregman divergence up to an additional term, independent of $f(x; \theta)$. See further discussion in App. B.

---

[1]Throughout, we simplify our notation by omitting the explicit dependence on $\lambda$ when possible. For example, we write $L(\mathscr{D}, \theta, w^n)$ instead of $L(\mathscr{D}, \theta, \lambda, w^n)$ whenever $\lambda = 0$.

## 2.1 Inference Objective

We study the *inference objective*, $T(\cdot, \cdot) : \mathbb{R}^d \times \mathbb{R}^n \to \mathbb{R}^k$, which maps a parameter vector $\theta$ and a weight vector $w^n$ to a desired inference target, where $w^n$ belongs to a family of weight vectors $\mathcal{W}$. In particular, we focus on cases where $w^n$ corresponds to a leave-one-out weight vector, defined as

$$\mathscr{D}^{-i} \triangleq \{w^n : w_j = \mathbb{1}\{j \neq i\}\}.$$

This formulation captures a range of tasks, including:

**Cross Validation**  To assess and select models, leave-one-out cross-validation (LOOCV) estimates model performance by iterative training on all but one data point and evaluating on the omitted instance. Specifically, for each $i$, it computes the evaluation metric:

$$T\left(\widehat{\theta}(w^n), w^n\right) \triangleq \frac{1}{n}\ell\left(z_i, \widehat{\theta}(w^n)\right) \quad \text{for} \quad w^n \in \mathscr{D}^{-i},$$

where $\mathscr{D}^{-i}$ denotes the leave-one-out weight vectors, $\widehat{\theta}(w^n)$ is the model trained with $w^n$, and the corresponding evaluation is taken on the omitted sample. Leave-$k$-out cross-validation follows analogously by choosing weights that correspond to removing a subset of $k$ observations (Geisser, 1975; Stone, 1974).

**Machine Unlearning**  To remove the influence of a data point $z_i$, the "unlearned model" is obtained by

$$T\left(\widehat{\theta}(w^n), w^n\right) = \widehat{\theta}(w^n) \quad \text{for} \quad w^n \in \mathscr{D}^{-i}.$$

This ensures that the model parameters are updated as if $z_i$ were never included in $\mathscr{D}$ (Cao and Yang, 2015). Similarly, unlearning $k$ data points follows the same formulation using leave-$k$-out weight vectors.

**Data attribution**  Understanding the contribution of a training sample $z_i \in \mathscr{D}$ to a model's prediction on a test point $z_{\text{test}}$ (Koh and Liang, 2017) is formulated as comparing $\ell\left(z_{\text{test}}, \widehat{\theta}(\vec{1})\right)$ with

$$T\left(\widehat{\theta}(w^n), w^n\right) = \ell\left(z_{\text{test}}, \widehat{\theta}(w^n)\right) \triangleq T\left(\widehat{\theta}(w^n)\right) \quad \text{for} \quad w^n \in \mathscr{D}^{-i}.$$

Attribution to a set of $k$ points follows analogously.

**Fairness Evaluation**  Recent works propose to evaluate the impact of $z_i$ on model fairness by computing $T\left(\widehat{\theta}(w^n)\right)$ for $w^n \in \mathscr{D}^{-i}$, where $T$ is a chosen fairness metric (Ghosh et al., 2023). For example, in a case one wants to protect a binary attribute represented by samples $\{s_i\}_{i=1}^n$, then an appropriate function $T\left(\widehat{\theta}(w^n)\right)$ can usually be defined as (see, for example, (Shah et al., 2024)):

$$T\left(\widehat{\theta}(w^n), w^n\right) \triangleq T\left(\widehat{\theta}(w^n)\right) \tag{2}$$

$$T\left(\widehat{\theta}(w^n)\right) = \left| \mathbb{E}_{\widehat{P}(\mathsf{x}|\mathsf{s}=0)}\left[f\left(\mathsf{x}; \widehat{\theta}(w^n)\right)\right] - \mathbb{E}_{\widehat{P}(\mathsf{x}|\mathsf{s}=1)}\left[f\left(\mathsf{x}; \widehat{\theta}(w^n)\right)\right] \right|, \quad \text{for } w^n \in \mathscr{D}^{-i}.$$

Here, $\widehat{P}(\mathsf{x} = x|\mathsf{s} = s)$ is the empirical distribution for $s \in \{0, 1\}$. Whenever the $\{s_i\}$ are continuous-valued, an alternative fairness metric can be defined via the $\chi^2$ divergence (Mary et al., 2019)

$$T\left(\widehat{\theta}(w^n), w^n\right) \triangleq T\left(\widehat{\theta}(w^n)\right) = \chi^2\left(\widehat{P}_{f(\mathsf{x};\widehat{\theta}(w^n)),\mathsf{s}} \| \widehat{P}_{f(\mathsf{x};\widehat{\theta}(w^n))}\widehat{P}_\mathsf{s}\right), \quad \text{for } w^n \in \mathscr{D}^{-i}.$$

The impact of removing a subset of $k$ samples is assessed analogously by considering $w^n \in \mathscr{D}^{-K}$.

*Remark* 2. Each application typically targets a distinct downstream task. Our aim, therefore, is to place them within a unified influence estimation framework rather than to develop task-specific formulations. Accordingly, we do not survey application-specific literature here and instead focus on the common formulation as an influence-estimation problem.

### 2.1.1 Inference Approximation

Since $\widehat{\theta}(w^n)$ for each weight vector is often computationally expensive, many methods approximate the inference objective using quantities derived from $\widehat{\theta}(\vec{1})$. That is, instead of solving for $\widehat{\theta}(w^n)$ directly, we use an approximation that combines the known vector $\widehat{\theta}(\vec{1})$ with a function of the weights $w^n$:

$$\widehat{\theta}(w^n) \approx g\left(\widehat{\theta}(\vec{1}), w^n\right) \triangleq \widetilde{\theta}(w^n).$$

Typically, $g(\cdot, \cdot)$ is derived from a Taylor series expansion around $\widehat{\theta}(\vec{1})$, capturing the $p$th-order sensitivity of the model parameters to small perturbations in $w^n$. Depending on $p$, this allows for efficient approximation without requiring full retraining (Giordano et al., 2019b;a; Wilson et al., 2020). Two widely used approaches to approximate the inference objective $T\left(\widehat{\theta}(w^n), w^n\right)$ are:

1. **Plug-in Estimator**: This approach directly substitutes $\widetilde{\theta}(w^n)$ into the inference objective:

$$T\left(\widehat{\theta}(w^n), w^n\right) \approx T\left(\widetilde{\theta}(w^n), w^n\right) = T\left(g\left(\widehat{\theta}(\vec{1}), w^n\right), w^n\right).$$

2. **Linearized Influence Approximation**: Instead of replacing $\widehat{\theta}(w^n)$ directly, this method uses a first-order expansion of $T\left(\widehat{\theta}(w^n), w^n\right)$ around $\widehat{\theta}(\vec{1})$. The approximation function $g(\cdot, \cdot)$ is then incorporated into this expansion to estimate $\widehat{\theta}(w^n)$:

$$T\left(\widehat{\theta}(w^n), w^n\right) \approx T\left(\widehat{\theta}(\vec{1}), w^n\right) + \left\langle \nabla_\theta T\left(\widehat{\theta}(\vec{1}), w^n\right), \widetilde{\theta}(w^n) - \widehat{\theta}(\vec{1}) \right\rangle. \tag{3}$$

Both methods are shown in several works to reduce computational overhead while performing well empirically (Koh and Liang, 2017; Koh et al., 2019; Wilson et al., 2020; Basu et al., 2020). However, the quality of the approximation depends on how well $g(\cdot, \cdot)$ captures the true parameter updates. In the next section, we introduce a new method for creating such an approximation.

## 3 Measuring Influence Using the Approximate Fisher Matrix

In this section, we provide background on measuring influence using the FIM. Starting from the classical Hessian-based formulation, we review why—and under what conditions—replacing the Hessian curvature with the FIM yields a viable surrogate for influence measurement, and we highlight the associated computational benefits. Our main contribution appears in Section. 4, where we build on this background to provide new accuracy guarantees for FIM-based influence estimators in a general setting that may involve non-differentiable regularizers.

A common approach to approximating $\hat{\theta}(w^n)$ is to optimize a surrogate to the loss function $L(\mathscr{D}, \theta, \lambda, w^n)$. This paper focuses on methods based on *quadratic approximations of the objective* (Cook and Weisberg, 1980; Koh and Liang, 2017; Giordano et al., 2019b; Wilson et al., 2020), which provide computationally efficient estimates while maintaining accuracy. These approximations yield solutions of the form:

$$\widetilde{\theta}(w^n) = \widehat{\theta}(\vec{1}) - \mathbf{C}\left(\widehat{\theta}(\vec{1}), w^n\right) b\left(\widehat{\theta}(\vec{1}), w^n\right),$$

where $b(\cdot, \cdot)$ and $\mathbf{C}(\cdot, \cdot)$ depend on the specific loss approximation and vary across applications.

A notable instance of this framework is the *infinitesimal jackknife* (IJ) approximation (Giordano et al., 2019b), denoted $\widetilde{\theta}^{\text{IJ}}(w^n)$, which is defined via a Newton step:

$$b\left(\widehat{\theta}(\vec{1}), w^n\right) \triangleq \frac{1}{n} \sum_{i=1}^{n} \nabla_\theta \ell\left(z_i, \widehat{\theta}(\vec{1})\right)(w_i - 1), \tag{4}$$

$$\mathbf{C}\left(\widehat{\theta}(\vec{1}), \vec{1}\right) = \nabla_\theta^2 L\left(\mathscr{D}, \widehat{\theta}(\vec{1}), \vec{1}\right)^{-1} \triangleq \mathbf{H}\left(\widehat{\theta}(\vec{1}), \vec{1}\right)^{-1}.$$

Extensions of these estimators have been studied under the more general framework where the loss $L$ contains a non-differentiable regularizer (Wilson et al., 2020; Suriyakumar and Wilson, 2022).

In this work, we examine a modified, computationally efficient approximation of $\widehat{\theta}(w^n)$ using the *natural gradient*. We consider functions $\ell$ that represent the negative log-likelihood of a parametric probabilistic model, $\ell(z, \theta) = -\log(P_{\mathsf{y}|\mathsf{x}}(y|f(x; \theta)))$, where $P_{\mathsf{y}|\mathsf{x}}(y|f(x; \theta))$ lies on the probability simplex of the output alphabet $\mathcal{Y}$ and parameterized by $\theta$ (Amari and Douglas, 1998; Amari, 2016). As discussed in (Banerjee et al., 2005) (see also App. B), this property holds for a large class of losses in machine learning. While the standard gradient identifies the direction that minimizes the objective based on Euclidean distance, the natural gradient accounts for the underlying geometry (curvature) of the parameter space. This is achieved by pre-multiplying the gradient with the inverse of the FIM, which characterizes the sensitivity of the model's likelihood function to changes in parameters. To that end, the Hessian in (4) is replaced by:

$$\mathbf{F}\left(\widehat{\theta}\left(\vec{1}\right)\right) \triangleq \mathbb{E}_{(\mathsf{x},\mathsf{y}) \sim P_{\mathsf{x},\mathsf{y}; \theta = \widehat{\theta}(\vec{1})}}[g_{\mathsf{y}|\mathsf{x}} \cdot g_{\mathsf{y}|\mathsf{x}}^{\top}]$$
$$g_{\mathsf{y}|\mathsf{x}} \triangleq \nabla_{\theta} \log\left(P_{\mathsf{y}|\mathsf{x}; \theta}\left(\mathsf{y}|\mathsf{x}; \widehat{\theta}(\vec{1})\right)\right),$$

where $P_{\mathsf{y}|\mathsf{x}}$ is the probabilistic model induced by the loss function (Martens, 2020). However, since the covariate distribution $P_{\mathsf{x}}$ is typically unknown, direct computation of the expectation is infeasible. Instead, we approximate the FIM using empirical estimates, averaging over the observed covariates and leveraging the network structure to evaluate expectations over $P_{\mathsf{y}|\mathsf{x}; \theta}$ (Martens, 2020; Kunstner et al., 2019). The resulting *approximate FIM* is given by:

$$\mathbf{F}\left(\mathscr{D}, \widehat{\theta}(\vec{1})\right) \triangleq \frac{1}{|\mathscr{D}|} \sum_{x \in \mathscr{D}} \mathbb{E}_{\mathsf{y} \sim P_{\mathsf{y}|\mathsf{x}=x; \theta = \widehat{\theta}(\vec{1})}}[g_{\mathsf{y}|\mathsf{x}} \cdot g_{\mathsf{y}|\mathsf{x}}^{\top}]. \tag{5}$$

Using this approximation, the *Approximate Fisher Infinitesimal Jackknife* is defined similarly to (4), replacing $\mathbf{C}$ with the approximate FIM $\mathbf{F}\left(\mathscr{D}, \widehat{\theta}(\vec{1})\right)$:

$$\tilde{\theta}^{\mathrm{IJ,AF}}(w^n) \triangleq \widehat{\theta}(\vec{1}) - \left(\mathbf{F}\left(\mathscr{D}, \widehat{\theta}(\vec{1})\right)\right)^{-1} b\left(\widehat{\theta}(\vec{1}), w^n\right). \tag{6}$$

Following classical results (e.g. (Schraudolph, 2002; Martens, 2020; Barshan et al., 2020)), when the loss function is given by $\ell(z, \theta) = -\log(P(y|f(x; \theta)))$ and $P(y|f)$ belongs to an exponential family, the approximate FIM can be interpreted as a positive semi-definite (PSD) approximation of the Hessian. Specifically, the Hessian satisfies:

$$\mathbf{H}\left(\widehat{\theta}(\vec{1}), \vec{1}\right) = \mathbf{F}\left(\mathscr{D}, \widehat{\theta}(\vec{1})\right) + \mathbf{R},$$

where $\mathbf{F}\left(\mathscr{D}, \widehat{\theta}(\vec{1})\right)$ is guaranteed to be PSD, and the remainder term is given by[2]:

$$\frac{1}{n} \sum_{i=1}^{n} \nabla_{\theta}^2 f\left(x_i; \widehat{\theta}(\vec{1})\right) \nabla_f \log\left(P\left(y_i|f\left(x_i; \widehat{\theta}(\vec{1})\right)\right)\right).$$

This remainder term can be non-zero, for example, in cases where $f(x; \theta)$ is non-linear in $\theta$. However, in many settings, including commonly used models, $\mathbf{R}$ shrinks to zero (in $L_2$ sense) as training accuracy improves (Kunstner et al., 2019; Martens, 2020) (see App. B, App. E). Thus, the approximated FIM is often viewed as a PSD approximation of the Hessian.

*Remark* 3. The definition of the approximated FIM (5) matches the definition from (Kunstner et al., 2019, Eq. 5). As discussed by Kunstner et al. (2019), some works call this quantity the empirical FIM. However, we adopt the terminology of (Kunstner et al., 2019), and refer to the empirical FIM as the quantity which uses the observed $y \in \mathscr{D}$ rather than taking expectation over $P_{\mathsf{y}|\mathsf{x}; \theta}$ (Kunstner et al., 2019, Eq. 6).

---

[2]Here "multiplication" means the mode-3 tensor–vector product: for $T \in \mathbb{R}^{\Theta \times \Theta \times |F|}$ and $v \in \mathbb{R}^{|F|}$, $T \times_3 v = \sum_{k=1}^{|F|} v_k T_{:,:,k}$.

*Remark* 4. The estimator $\tilde{\theta}^{\text{IJ,AF}}(w^n)$ has been considered as an alternative to the jackknife influence estimator $\tilde{\theta}^{\text{IJ}}(w^n)$ in multiple different works, for example (Bae et al., 2022; Barshan et al., 2020; Teso et al., 2021). However, as we show in Section. 4, our new analysis yields improved accuracy guarantees for such estimators in a more general setting that allows non-differentiable regularizers.

## 3.1 Computational Efficiency By Using the FIM

One advantage of using the FIM instead of the Hessian is its more favorable computational efficiency, which we demonstrate next. Both the Hessian and the FIM-based estimators require computing expressions of the form $\mathbf{A}^{-1}b\left(\widehat{\theta}(\vec{1}), w^n\right)$ where $\mathbf{A} = \mathbf{F}\left(\mathscr{D}, \widehat{\theta}(\vec{1})\right)$ in (9) and $\mathbf{A} = \mathbf{H}\left(\widehat{\theta}(\vec{1}), \vec{1}\right)$ for the IJ. Since directly inverting a large $d \times d$ matrix is usually infeasible, efficient computation of inverse-matrix-vector products is essential. However, since the FIM requires only first-order differentiation through the model, it will typically be much more computationally efficient relative to the Hessian-based alternative. We demonstrate this property by analyzing the application of the FIM with the `LiSSA` algorithm (Agarwal et al., 2017). Similar efficiency arguments extend to modern variants of influence methods that rely on stochastic inverse-matrix-vector product estimation, such as (Guo et al., 2021; Schioppa et al., 2022).

### 3.1.1 Stochastic Estimation

Stochastic estimation techniques rely on generating a sequence of estimators $v_j \triangleq \widehat{(\mathbf{A}^{-1}x)}_j$ that converge in expectation to $\mathbf{A}^{-1}x$ as $j \to \infty$, where each $v_j$ utilizes only a small batch of training data, yielding a computationally tractable way to estimate $\mathbf{A}^{-1}x$. As an example of the computational superiority of the FIM-based methods, we will demonstrate the improvement for the celebrated `LiSSA` algorithm (Agarwal et al., 2017), though similar arguments holds for most automatic differentiation techniques used for carrying out such calculations. The `LiSSA` algorithm approximates $\mathbf{A}^{-1}$ using the truncated Neumann series $\mathbf{A}_j^{-1} = \sum_{i=0}^{j}(\mathbb{I} - \sigma\mathbf{A})^i$ for some $\sigma > 0$ [3], which we further note can be equivalently written via the recursion $\mathbf{A}_j^{-1} = \mathbb{I} + (\mathbb{I} - \sigma\mathbf{A})\mathbf{A}_{j-1}^{-1}$. Consequently, each $v_j$ is defined by $v_j = x + (\mathbb{I} - \sigma\mathbf{A})v_{j-1}$ with $v_0 = x$ and final estimate $v = \sigma v_N$. The major computational hurdle is multiplying by $\mathbf{A}$. When $\mathbf{A}$ depends on many training points $\left(\text{e.g.,} \mathbf{F}\left(\mathscr{D}, \widehat{\theta}(\vec{1})\right) \text{ or } \mathbf{H}\left(\widehat{\theta}(\vec{1}), \vec{1}\right)\right)$, it is typical to estimate it by using a sampled batch of training data. We now analyze the computational complexity of these calculations for each method.

**Estimation with $\mathbf{F}\left(\mathscr{D}, \widehat{\theta}(\vec{1})\right)$.** When $\mathbf{A} \triangleq \mathbf{F}(\mathscr{D}, \theta)$, each $\mathbf{A}v_j$ requires calculating

$$\nabla_\theta f_i \cdot \left(\nabla_f^2 \log(P(y_i|f_i))\right)\left(v_j^\top \nabla_\theta f_i\right)^\top \tag{7}$$

where $f_i \triangleq f(x_i; \theta)$. Given the form of $\nabla_f^2 \log(P(y_j|f(x_j; \theta)))$ (see App. B), computing (7) requires the vector-Jacobian product (VJP) $a_j = v_j^\top \nabla_\theta f_i$ and the Jacobian-vector product (JVP) $\nabla_\theta f_i \cdot \left(\nabla_f^2 \log(P(y_i|f_i))\right)a_j^\top$.

**Estimation with $\mathbf{H}\left(\widehat{\theta}(\vec{1}), \vec{1}\right)$.** For $\mathbf{A} \triangleq \mathbf{H}\left(\widehat{\theta}(\vec{1}), \vec{1}\right)$, each $\mathbf{A}v_j$ requires computing,

$$\nabla_\theta^2 \log(P(y_i|f_i))v_j, \tag{8}$$

which requires computing a Hessian-vector product (HVP) with respect to all model parameters.

**Comparing Computations.** Computing (8) requires roughly four evaluations of the entire model (Schraudolph, 2002; Dagréou et al., 2024). In contrast, a JVP can be computed in a single forward pass using forward-mode automatic differentiation (Bradbury et al., 2018). Since $\nabla_f^2 \log(P(y_i|f_i))$ is typically simple and depends only on the number of model outputs (not on $d$), evaluating (7) requires just one differentiation in backward mode that happens inside the VJP. We note that this operation is the only computationally heavy backward differentiation, which usually requires one forward pass followed by a backward pass (Dagréou et al., 2024), in contrast to direct HVP evaluation, which usually requires two backward differentiations.

---

[3]$\sigma$ is usually a small positive constant to stabilize calculations.

Furthermore, given backward differentiation roughly costs twice forward passes (DeepSpeed, 2024; Dagréou et al., 2024), this method significantly reduces FLOPs and accelerates computations. We demonstrate these savings through simulations in Section. 5, and summarize the results in Tbl. 1.

| | Forward | Backward | FLOPs |
|---|---|---|---|
| (8) | 0 | 2 | $O(4F)$ |
| (8) | 2 | 1 | $O(4F)$ |
| (7) | 1 | 1 | $O(3F)$ |

Table 1: Number of differentiations in forward mode, backward mode, and FLOPs required to evaluate (7) and (8), for different evaluation options from (Dagréou et al., 2024). $F$ denotes the FLOPs needed for a single model forward pass.

*Remark* 5. Although our analysis focuses on the `LiSSA` algorithm, for which implementations with the approximated FIM have been previously explored (Bae et al., 2022), the fact that the FIM depends solely on first-order gradients means these improvements are broadly applicable to many methods that require differentiating through a large model using the structure of the curvature matrix. For example, similar fundamental gains were observed in (Sattigeri et al., 2022) by employing efficient matrix-inversion techniques based on rank-one updates.

*Remark* 6. The classical `LiSSA` method has been refined into more scalable variants (e.g., (Guo et al., 2021)), but these approaches still rely on computing matrix–vector products. Replacing the Hessian with the FIM in such implementations yields similar computational savings as in the standard `LiSSA` setting we show here.

## 4 Theoretical Analysis

Following Section. 3, since the FIM behaves as a viable replacement of the Hessian, which is also usually more computationally efficient, it is necessary to have an accuracy analysis for tasks that use it for influence measurement. To that end, we now present our new theoretical analysis, where we provide guarantees on the accuracy of a general influence estimator that is based on the FIM. We then demonstrate the applications of these guarantees for multiple different problems involving influence estimation. While similar results are well understood for infinitesimal jackknife-based approximations (Giordano et al., 2019b; Wilson et al., 2020; Suriyakumar and Wilson, 2022; Sekhari et al., 2021), our framework extends these findings to also cover settings when one replaces the Hessian with the approximated FIM and further adds a non-differentiable regularizer, showing that, in general, (6) can be safely used in some places currently invoking (4).

### 4.1 Related work

Several works have established the accuracy of this approximation under specific conditions on the loss function and the weight vectors $w^n$ (Giordano et al., 2019b; Wilson et al., 2020; Suriyakumar and Wilson, 2022; Sekhari et al., 2021). These results hold under subsets of the following assumptions.

*Assumption* 1 (Curvature of the Objective). For each $i \in [n]$, the function $\frac{1}{n}\ell(z_i, \theta)$ is $\mu$-strongly convex ($\mu > 0$), and the regularizer $\pi(\theta)$ is convex.

*Assumption* 2 (Lipschitz Hessian of the Objective). For each $i \in [n]$, the function $\frac{1}{n}\ell(z_i, \theta)$ is twice differentiable with an $M$-Lipschitz Hessian.

*Assumption* 3 (Smooth Hessian of the Objective). For each $i \in [n]$, the function $\frac{1}{n}\ell(z_i, \theta)$ is twice differentiable with a $C$-smooth Hessian.

*Assumption* 4 (Bounded Moments). For given $s, r \geq 0$, the quantity $B_{sr}$ is finite, where

$$B_{sr} \triangleq \frac{1}{n}\sum_{i=1}^{n} \text{Lip}\left(\nabla_\theta \ell(z_i, \cdot)\right)^s \left\|\nabla_\theta \ell\left(z_i, \widehat{\theta}(\vec{1})\right)\right\|^r.$$

*Assumption* 5 (Lipschitz Features). The feature mapping $f(x_i; \theta)$ is $C_f$-Lipschitz with a $\widetilde{C}_f$-Lipschitz gradient for all $i \in [n]$.

*Assumption* 6 (Lipschitz Inference Objective)**.** The inference objective $T(\theta, w^n)$ is twice differentiable, $C_{T_1}$-Lipschitz, and has a $C_{T_2}$-Lipschitz gradient with respect to $\theta$ for $w^n \in \mathscr{D}^{-i}$ and all $i \in [n]$.

For the examples in Section. 2, the following guarantees hold under subsets of Assumption. 1–Assumption. 6:

**Proposition 1** (LOOCV Approximation Bound ((Wilson et al., 2020), Thm. 4))**.** *Suppose Assumption. 1, Assumption. 2, and Assumption. 4 hold for $(s, r) = \{(0, 3), (1, 3), (1, 4), (1, 2), (2, 2), (3, 2)\}$. When the IJ is used as a plug-in estimate for the LOOCV objective*

$$T_i(\theta) \triangleq T(\theta, w^n) = \frac{1}{n} \ell(z_i, \theta),$$

*with $w^n \in \mathscr{D}^{-i}$, the error in this approximation is bounded as*

$$\left| \sum_{i=1}^n \left( T_i\left(\widetilde{\theta}^{\mathrm{IJ}}(w^n)\right) - T_i\left(\widehat{\theta}(w^n)\right) \right) \right| = O\left( \frac{MB_{03}}{\mu^3 n^2} + \frac{B_{12}}{\mu^2 n^2} \right).$$

The next proposition relies on the $(\varepsilon, \delta)$-unlearning definition from (Sekhari et al., 2021).

**Proposition 2** (Machine Unlearning (Suriyakumar and Wilson, 2022))**.** *Suppose $\ell(z, \theta)$ is $\mu$-strongly convex, twice differentiable, $L$-Lipschitz, with a $C$-smooth and $M$-Lipschitz Hessian for all $z$, and that $\pi(\theta)$ is convex.[4] When the IJ is used as a plug-in estimate for the objective $T(\theta) = \theta$, we have*

$$\left\| T\left(\widetilde{\theta}^{\mathrm{IJ}}(w^n)\right) - T\left(\widehat{\theta}(w^n)\right) \right\| \leq \frac{2ML}{n^2\mu^2} + \frac{CL^2}{n^2\mu^3}, \quad for\ w^n \in \mathscr{D}^{-i}.$$

*Furthermore, the algorithm returning $\widetilde{\theta}^{\mathrm{IJ}}(w^n) + \zeta$ for $w^n \in \mathscr{D}^{-i}$ satisfies $(\varepsilon, \delta)$-unlearning, where $\zeta \sim \mathcal{N}(0, c\mathbb{I})$ with $c = (2\mu ML + CL^2) \frac{\sqrt{2\log(5/4\delta)}}{\varepsilon \mu^3 n^2}$.*

**Proposition 3** (Data Attribution ((Koh et al., 2019), Prop. 1))**.** *Suppose Assumption. 1, Assumption. 2, and Assumption. 6 hold, and that $\pi(\theta) = \|\theta\|^2$. Define $C_\ell \triangleq \max_{i \in [n]} \left\| \nabla \ell\left(z_i, \widehat{\theta}(\vec{1})\right) \right\|$. When the IJ is used as a plug-in estimate for the inference objective*

$$T(\theta) = \ell(z_{\text{test}}, \theta) - \ell\left(z_{\text{test}}, \widehat{\theta}(\vec{1})\right),$$

*the approximation error is bounded as*

$$\left| T\left(\widehat{\theta}(w^n)\right) - T\left(\widetilde{\theta}^{\mathrm{IJ}}(w^n)\right) \right| \leq \frac{MC_{T_1}C_\ell^2}{n^2\mu^3}, \quad for\ w^n \in \mathscr{D}^{-i}.$$

While certain loss functions may not be Lipschitz, Assumption. 2 and Assumption. 4 require only that the *normalized* losses evaluated on the training set satisfy Lipschitz continuity— a condition that generally holds in practice (Giordano et al., 2019b, Assump. 3). Similarly, when the inference objective is of the form $\ell(z_{\text{test}}, \theta)$, Lipschitz continuity is required only with respect to the test point $z_{\text{test}}$. As long as $z_{\text{test}}$ is not pathological, this assumption is typically satisfied.[5]

Additionally, the framework in (Giordano et al., 2019b) assumes differentiable regularization. In certain cases, similar approximations extend to settings where the regularizer is non-differentiable (Wilson et al., 2020; Suriyakumar and Wilson, 2022).

### 4.2 The Approximate Fisher Influence Function

We now present an influence estimator that is based on the FIM and its accuracy guarantees. As mentioned, our estimator accommodates non-differentiable regularizers, and our analysis provides new theoretical guarantees for influence measurement using the FIM. First, we introduce an additional technical assumption about the loss function, which is essential for our proofs.

---

[4]These assumptions strengthen Assumption. 1-Assumption. 4, requiring Lipschitz continuity for any $z$, not just the training samples $\{z_i\}$.

[5]The assumption that $T$ is Lipschitz is consistent with classical works on influence functions; see (Koh et al., 2019, Prop. 1).

*Assumption* 7. The loss functions are of the form $\ell(z, \theta) = -\log(P(y|f(x; \theta)))$ where $P(y|f)$ belongs to a regular exponential family whose natural parameters are $f(x; \theta)$. Moreover, we further assume that $\left\| \nabla_f^2 \log(P(y|f(x; \theta))) \right\| \leq Q$ for some $Q > 0$.

To accommodate non-smooth regularizers, we utilize the *proximal operator*, defined as:

$$\text{prox}_{\lambda\pi}^{\mathbf{D}}(v) \triangleq \underset{\theta}{\text{argmin}} \ \left\{ (v - \theta)^\top \mathbf{D}(v - \theta) + 2\lambda\pi(\theta) \right\}.$$

We note that the usage of the proximal operator in influence measurement tasks has been done previously in the context of cross-validation (Wilson et al., 2020) and machine unlearning (Suriyakumar and Wilson, 2022). Our formulation below extends this to settings that use the FIM instead of the Hessian.

The following lemma defines the approximate Fisher influence and bounds the distance to $\widehat{\theta}(w^n)$ for $w^n \in \mathscr{D}^{-i}$.

**Lemma 1.** *Suppose Assumption. 1, Assumption. 2, Assumption. 5, and Assumption. 7 hold. Define* $\bar{E}_n \triangleq \sum_{j=1}^n \left| \nabla_f \log\left( P\left( y_j | f\left( x_j; \widehat{\theta}(\vec{1}) \right) \right) \right) \right|$, $\widetilde{g}_i \triangleq \left| \nabla_\theta \ell\left( z_i, \widehat{\theta}(\vec{1}) \right) \right|$ *and* $g_i = \frac{\widetilde{g}_i}{n}$. *Then, the approximated Fisher influence function, defined via*

$$\widetilde{\theta}(w^n) = \text{Prox}_{\lambda\pi}^{\mathbf{F}\left( \mathscr{D}, \widehat{\theta}(\vec{1}) \right)} \left( \widetilde{\theta}^{\text{IJ,AF}}(w^n) \right), \quad \text{for } w^n \in \mathscr{D}^{-i}. \tag{9}$$

*satisfies*

$$\left\| \widetilde{\theta}(w^n) - \widehat{\theta}(w^n) \right\| \leq \frac{2QC_f^2 \widetilde{g}_i}{n^2\mu^2} + \frac{M\widetilde{g}_i^2}{n^2\mu^3} + \frac{2\widetilde{g}_i \widetilde{C}_f \bar{E}_n}{n\mu^2}, \quad \text{for } w^n \in \mathscr{D}^{-i}. \tag{10}$$

*Proof sketch.* The proof separately bounds the distances between (i) $\widehat{\theta}(w^n)$ and $\widetilde{\theta}^{\text{IJ}}(w^n)$, and (ii) $\widetilde{\theta}^{\text{IJ}}(w^n)$ and $\widetilde{\theta}^{\text{IJ,AF}}(w^n)$ for $w^n \in \mathscr{D}^{-i}$. The first bound follows from (Wilson et al., 2020, Lem. 1), while the second leverages the closeness of the Hessian and the FIM. Full proof is provided in App. E. □

Similar to prior results (Wilson et al., 2020; Suriyakumar and Wilson, 2022; Sekhari et al., 2021), the first two terms in (10) depend on global problem constants (Lipschitz coefficients, strong convexity parameter, etc.) and the gradient at the $i$th training point. The third term depends on $\bar{E}_n$, which simplifies due to the exponential family structure of the loss and is given by (see App. G)

$$\left\| \nabla_f \log\left( P\left( y_i | f\left( x_i; \widehat{\theta}(\vec{1}) \right) \right) \right) \right\| = \left| t(y_i) - \mathbb{E}_{\mathsf{y} \sim P_{\mathsf{y}|\mathsf{x} = x_i; \widehat{\theta}(\vec{1})}} [t(\mathsf{y})] \right|.$$

Moreover, $\bar{E}_n$ can be shown to serve as an upper bound on the gradient at the optimum $\widehat{\theta}(\vec{1})$ (see App. B, App. C). In App. B, we further demonstrate how this term relates to the absolute training error in classification and regression problems. Specifically, as training error decreases, this term also diminishes. In the extreme case where $\ell\left( z_i, \widehat{\theta}(\vec{1}) \right) = 0$ for all $i \in [n]$, this term is exactly zero (see App. E). Thus, we expect the excess term in (10) to be small whenever the model's training loss is small. For the remaining terms in Lemma. 1, the worst-case discrepancy between $\widetilde{\theta}(w^n)$ and $\widehat{\theta}(w^n)$ for all $i \in [n]$ is controlled by $g_{\max} \triangleq \max_{i \in [n]} g_i$. By Assumption. 4 with $(s, r) = (0, 1)$, $g_{\max}$ is finite. Next, we present our main theorem, which establishes error bounds for the approximated inference objective $T(\cdot)$.

**Theorem 1.** *Suppose Assumption. 1, Assumption. 2, and Assumption. 5-Assumption. 7 hold. Let* $\widetilde{\theta}(w^n)$ *be defined as in (9) for* $w^n \in \mathscr{D}^{-i}$. *Then,*

$$\left\| T\left( \widehat{\theta}(w^n) \right) - T\left( \widetilde{\theta}(w^n) \right) \right\| \leq C_{T_1} \left( \frac{2QC_f^2 \widetilde{g}_i}{n^2\mu^2} + \frac{M\widetilde{g}_i^2}{n^2\mu^3} + \frac{2\widetilde{g}_i \widetilde{C}_f \bar{E}_n}{n\mu^2} \right) \tag{11}$$

$$+ \frac{1}{2} C_{T_2} \left( \frac{2QC_f^2 \widetilde{g}_i}{n^2\mu^2} + \frac{M\widetilde{g}_i^2}{n^2\mu^3} + \frac{2\widetilde{g}_i \widetilde{C}_f \bar{E}_n}{n\mu^2} \right)^2$$

*and,*

$$\left\| T\left(\widehat{\theta}\left(w^n\right)\right) - T\left(\widehat{\theta}(\vec{1})\right) - \left\langle \nabla T\left(\widehat{\theta}(\vec{1})\right), \widetilde{\theta}(w^n) - \widehat{\theta}(\vec{1})\right\rangle \right\| \tag{12}$$
$$\leq C_{T_1}\left(\frac{2QC_f^2\tilde{g}_i}{n^2\mu^2} + \frac{M\tilde{g}_i^2}{n^2\mu^3} + \frac{2\tilde{g}_i\widetilde{C}_f\bar{E}_n}{n\mu^2}\right) + \frac{2C_{T_2}\tilde{g}_i^2}{n^2\mu^2}.$$

*Proof sketch.* Both bounds follow from the smoothness properties of $T$ (Assumption. 6), combined with Lemma. 1 and Lemma. 2 from App. D. Full proof is provided in App. H. □

Theorem. 1 enables a systematic derivation of theoretical guarantees for FIM-based influence approximations across various application areas. Moreover, as discussed in (Giordano et al., 2019b, Sec. 3), for weight vectors $w^n = \mathscr{D}^{-i}$, we expect $\lim_{n\to\infty} g_{\max} = 0$. Consequently, whenever $\bar{E}_n = O(n^{-1})$, Theorem. 1 ensures that $T\left(\widetilde{\theta}(w^n)\right)$ and the Taylor-series approximation (Equation (3) with $w^n \in \mathscr{D}^{-i}$) converge to $T\left(\widehat{\theta}\left(w^n\right)\right)$ for all $i \in [n]$. However, as we demonstrate in Section. 5, in practice, $\widetilde{\theta}(w^n)$ is often a good approximation of $\widehat{\theta}\left(w^n\right)$ even when $\bar{E}_n$ is finite.

Next, building on Theorem. 1 and by leveraging influence function techniques established in prior works (Giordano et al., 2019a; Wilson et al., 2020; Sekhari et al., 2021; Suriyakumar and Wilson, 2022), we derive results analogous to those presented in Proposition. 1 through Proposition. 3.

**Corollary 1** (LOOCV). *Suppose Assumption. 1, Assumption. 2, and Assumption. 4-Assumption. 7 hold with $(s, r) = \{(0, 2), (0, 3), (1, 2), (1, 3), (1, 4)\}$. Let $T\left(\theta, \vec{1}^{n\backslash i}\right) = \frac{1}{n}\ell(z_i, \theta) \triangleq T_i(\theta)$. When $\widetilde{\theta}(w^n)$ from Lemma. 1 is used as a plug-in estimate for $w^n \in \mathscr{D}^{-i}$, the error in the approximate cross-validation estimate satisfies:*

$$\left|\sum_{i=1}^n \left(T_i\left(\widetilde{\theta}\left(\vec{1}^{n\backslash i}\right)\right) - T_i\left(\widehat{\theta}(\vec{1}^{n\backslash i})\right)\right)\right| \leq O\left(\frac{MB_{03}}{\mu^3 n^2} + \frac{C_f^2 B_{02}}{\mu^2 n^2} + \frac{\widetilde{C}_f\bar{E}_n B_{02}}{\mu^2 n}\right).$$

**Corollary 2** (Machine Unlearning). *Suppose Assumption. 1, Assumption. 2, Assumption. 5, and Assumption. 7 hold. Assume that $\widetilde{g}_i \leq G$ for all $i \in [n]$. Then, for the inference objective $T(\theta) = \theta$, we have:*

$$\left\| T\left(\widetilde{\theta}(w^n)\right) - T\left(\widehat{\theta}\left(w^n\right)\right)\right\| \leq \frac{2QC_f^2 G}{n^2\mu^2} + \frac{MG^2}{n^2\mu^3} + \frac{2G\widetilde{C}_f\bar{E}_n}{n\mu^2}, \quad \text{for } w^n \in \mathscr{D}^{-i}.$$

*Furthermore, the algorithm returning $\widetilde{\theta}(w^n) + \zeta$ satisfies $(\varepsilon, \delta)$-unlearning, where $\zeta \sim \mathcal{N}(0, c\mathbb{I})$ and:*

$$c = \left(\frac{2QC_f^2 G}{n^2\mu^2} + \frac{MG^2}{n^2\mu^3} + \frac{2G\widetilde{C}_f\bar{E}_n}{n\mu^2}\right)\frac{\sqrt{2\log(5/4\delta)}}{\varepsilon}.$$

**Corollary 3** (Data Attribution). *Suppose the assumptions of Theorem. 1 hold, $T(\theta) = \ell(z_{\text{test}}, \theta) - \ell\left(z_{\text{test}}, \widehat{\theta}(\vec{1})\right)$ and $C_\ell \triangleq \max_{i\in[n]} \widetilde{g}_i$. Then,*

$$\left| T\left(\widehat{\theta}\left(\vec{1}^{n\backslash i}\right)\right) - T\left(\widetilde{\theta}\left(\vec{1}^{n\backslash i}\right)\right)\right| \leq O\left(\frac{C_f^2 C_{T_1} C_\ell}{n^2\mu^2} + \frac{MC_{T_1} C_\ell^2}{n^2\mu^3} + \frac{C_{T_1}\widetilde{C}_f\bar{E}_n C_\ell}{n\mu^2}\right),$$

$$\left| T\left(\widehat{\theta}\left(\vec{1}^{n\backslash i}\right)\right) - T\left(\widehat{\theta}(\vec{1})\right) - \left\langle \nabla T\left(\widehat{\theta}(\vec{1})\right), \widetilde{\theta}\left(\vec{1}^{n\backslash i}\right) - \widehat{\theta}(\vec{1})\right\rangle\right|$$
$$\leq O\left(\frac{C_f^2 C_{T_1} C_\ell}{n^2\mu^2} + \frac{MC_{T_1} C_\ell^2}{n^2\mu^3} + \frac{C_{T_2} C_\ell^2}{n^2\mu^2} + \frac{C_{T_1}\widetilde{C}_f\bar{E}_n C_\ell}{n\mu^2}\right)$$

The proofs for these corollaries rely on applying Theorem. 1 for the settings described in Proposition. 1 - Proposition. 3 and follow via similar analysis techniques as presented in the prior works (Giordano et al., 2019b; Wilson et al., 2020; Sekhari et al., 2021; Suriyakumar and Wilson, 2022) (see App. I). We further provide guarantees for the fairness assessment task described in Section. 2, for which currently there is no theoretical analysis. The proof is in App. I.4.

**Corollary 4** (Fairness Evaluation)**.** *Suppose Assumption. 1, Assumption. 2, and Assumption. 5 - Assumption. 7 hold. If $T$ be given by* (2) *and* $C_\ell \triangleq \max\limits_{i \in [n]} \widetilde{g}_i$. *Then,*

$$\left| T\left(\widehat{\theta}\left(\vec{1}^{n \backslash i}\right)\right) - T\left(\widetilde{\theta}\left(\vec{1}^{n \backslash i}\right)\right)\right| \leq O\left(\frac{C_f^3 C_\ell}{n^2 \mu^2} + \frac{M C_f C_\ell^2}{n^2 \mu^3} + \frac{C_f \widetilde{C}_f C_\ell \bar{E}_n}{n \mu^2}\right).$$

We note that this formulation further extends to additional problems in machine learning and statistics beyond the specific applications discussed (e.g., data dropping (Broderick et al., 2020)).

*Remark* 7 (Practicality of Technical Assumptions)*.* Our analysis relies on standard regularity conditions on the per-sample losses $\{\ell(z_i; \theta)\}$, the functions $\{f(x_i; \theta)\}$, and the objective $T(\theta, w^n)$. These are broadly consistent with those used in closely related work (e.g., (Giordano et al., 2019b; Sekhari et al., 2021; Koh et al., 2019)). In particular, Assumption. 1–Assumption. 3 mirror Assumptions 1–4 of Giordano et al. (2019b) for the data-fit term (i.e., the loss excluding $\pi$), while Assumption. 4 is analogous to the bounded-moment condition in Wilson et al. (2020, Assumption 2). Moreover, Assumption. 6 matches the Lipschitz/smoothness requirements that are standard in classical influence-function analyses (e.g., (Koh et al., 2019)), and Assumption. 7 is satisfied by many common objectives functions in machine learning. Finally, Assumption. 5 typically holds in many practical settings (see, for example, (Virmaux and Scaman, 2018)).

*Remark* 8 (Non-Differentiable Regularizers)*.* Our framework allows for a general regularizer term (not necessarily differentiable), and can be calculated efficiently as long as its proximal operator can be computed easily. Since training models with general regularization (beyond $L_2$) is an increasingly popular method for adding robustness, feature sparsity, and interoperability to models (see (Lemhadri et al., 2021; Li et al., 2021) and references therein), we see this as a major advantage of our method. Moreover, we note that this differentiates our method from previous works that considered the FIM in influence estimation tasks, such as (Bae et al., 2022; Choe et al., 2024; Park et al., 2023). Furthermore, as mentioned in (Wilson et al., 2020; Suriyakumar and Wilson, 2022), since only a single step of the proximal operator is needed, its calculation is feasible and can be made computationally efficient in many practical applications. Thus, within a similar framework of influence estimation, the inclusion of this proximal operator is not expected to add any additional substantial computational complexity upon the currently existing schemes.

*Remark* 9 (General Non-Convex Setting)*.* While the Hessian of a non-convex objective can be indefinite, the Fisher Information Matrix (FIM) is positive semidefinite by construction. This eliminates negative-curvature directions and typically yields more stable curvature solves, which in turn, as observed in multiple prior works, improves the numerical robustness of influence estimates in practice (Martens and Grosse, 2015; Bae et al., 2022; Park et al., 2023). Our theoretical results complement these empirical findings by characterizing conditions under which replacing the Hessian curvature with the FIM preserves influence accuracy.

## 5 Experiments

We evaluate the utility of the approximate Fisher influence through experiments of two different settings. Both Fisher-based and Hessian-based influence functions are implemented within the same codebase, differing only in the automatic differentiation components used to compute (7) and (8). We first validate our theoretical framework by showing that whenever the technical conditions are met, FIM-based influence estimates are nearly indistinguishable from Hessian-based ones, despite being significantly faster to compute. We then transition to practical scenarios that challenge our theoretical assumptions, where we show that the influence estimator with the FIM might also outperform estimators that use the Hessian, while running faster. Both cases simulate scenarios where a non-differentiable regularizer is used. Detailed experimental procedures are provided in App. J. Our objective is to demonstrate the advantages of AFIF across different tasks, as reflected by our analysis, namely, to show that:

1. Within the theoretical framework we analyze, it achieves similar utility as the Hessian-based techniques;

2. It can be computed more efficiently relative to the Hessian-based techniques;

3. It reliably estimates influence even in cases with non-differentiable regularizers.

### 5.1 Logistic Regression With Non-Linear Natural Parameter Map

Our first experiment considers a logistic regression objective with a non-linear natural-parameter map. Given data $\{(x_i, y_i)\}_{i=1}^n$ with $x_i \in \mathbb{R}^d$ and $y_i \in \{0, 1\}$, we fit

$$\widehat{\theta} = \underset{\theta \in \mathbb{R}^p}{\operatorname{argmin}} \left\{ \frac{1}{n} \sum_{i=1}^n \left( \log\left(1 + e^{\eta_i}\right) - y_i \eta_i \right) + \lambda \|\theta\|_1 \right\}, \qquad \eta_i = \theta^\top \phi(x_i) + \frac{\alpha}{2} \|\theta\|_2^2, \tag{13}$$

where $\phi(\cdot) : \mathbb{R}^d \to \mathbb{R}^p$ is a fixed feature map (implemented via random Fourier features (Rahimi and Recht, 2007)), and $\alpha > 0$ introduces a non-linearity in the parameter-to-logit map. We generate the base vector $\theta$ such that only 10% of its entries are non-zero. As shown in App. J, the per-sample loss is the negative log-likelihood of a regular exponential family with uniformly bounded curvature in the natural parameter, and the regularization term ensures strong convexity. Our goal is to estimate the $K$-fold CV loss as a function of $\lambda$. For each split, we approximate the leave-fold-out refit using influence functions, where the required linear solver takes the form $\mathbf{H}^{-1}v$ (Hessian-based) or $\mathbf{F}^{-1}v$ (FIM-based). Rather than forming or inverting these matrices explicitly, we compute these products using the conjugate gradient method with Hessian-vector products and Fisher-vector products obtained via automatic differentiation (see App. J for details). As shown in Figure. 1, the Hessian- and FIM-based influence estimators yield nearly identical CV loss estimates across $\lambda$. Moreover, the FIM-based estimator is consistently faster to compute.

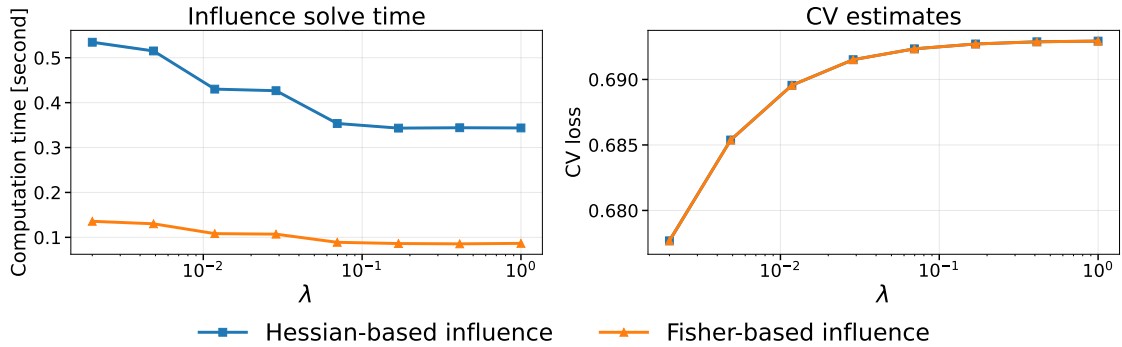

Figure 1: CV loss estimation for logistic regression with non-linear natural parameter map

### 5.2 Cross-Validation

In our second experiment, we evaluate both the computational cost and the accuracy of our influence-based CV approximation. We use a two-layer network on the Friedman-1 dataset (Friedman, 1991) and track performance across training epochs. To facilitate the need for $\ell_1$ regularization, we have artificially added random features to the training set, such that a feature selection is needed for optimal performance, and we thus trained the model with $\ell_1$ regularization. To approximate LOOCV efficiently, we use five-fold CV and apply our FIM-based update (Corollary. 1, adapted to leave-$k$-out) and also a variant that uses the Hessian instead of the FIM. For each epoch we report: (i) the held-out test loss, (ii) exact five-fold CV (obtained by retraining the model on a subset of the training set and testing it on an held-out validation set), (iii) our FIM-based approximate CV and an Hessian-based approximate CV, and (iv) the average wall-clock time to compute each CV quantity across the five folds. As is clear from Figure. 2, our influence measurement technique allows for a reliable CV approximation and at a substantially reduced computation time, supporting the claim that our method gives an option for measuring influence in situations where one uses non-differentiable regularizers, and that further our framework generalizes across different settings.

## 6 Concluding Remarks and Future Research

In this work, we introduced AFIF, a framework for influence estimation in statistical models based on the FIM. We showed that, across several classical influence-measurement tasks, the FIM can replace the Hessian

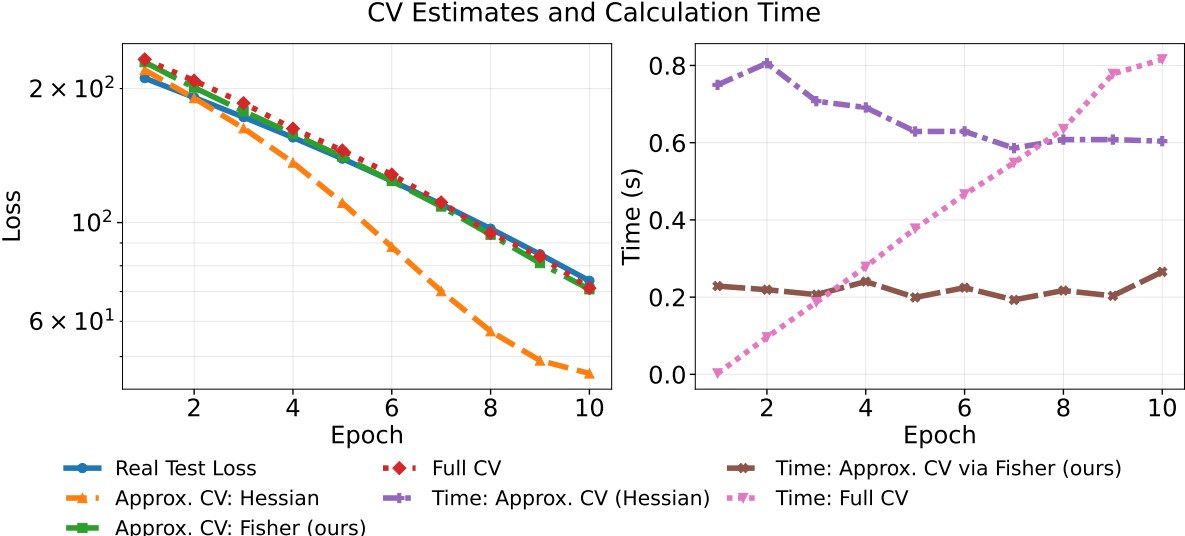

Figure 2: Held-out test loss, exact five-fold CV, FIM-based approximate CV (Corollary. 1 adapted to leave-$k$-out), and average per-epoch compute time (averaged across folds) for a two-layer network on the Friedman-1 dataset (Friedman, 1991). The time represents the average accumulated time to create one CV estimate for a certain number of epochs.

while providing comparable error guarantees and lower computational cost. In addition, AFIF extends naturally to settings involving non-differentiable regularizers, thereby expanding the applicability of influence estimation beyond the standard setting. Our experiments demonstrate that AFIF yields reliable influence estimates across a range of tasks, achieves runtime improvements relative to Hessian-based baselines, and remains effective even in practical settings that depart from the assumptions of our theory.

An important direction for future work is to extend this analysis to richer FIM-based influence methods used in practice, such as Kronecker-factored approximations of the FIM (Choe et al., 2024), which currently lack comparable theoretical guarantees. Another promising direction is to develop computationally efficient higher-order influence methods (Giordano et al., 2019a; Basu et al., 2020) by further leveraging the statistical structure of the learning problem.

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

# A    Definitions and Useful Lemmas

In our paper we use the next classical definitions from the convex optimization theory (Nesterov, 2013).

**Definition 1** (Matrix Operator-Norm)**.** For any matrix $\mathbf{A}$ we define its *operator-norm* by $\|\mathbf{A}\|_{\mathrm{op}} \triangleq \sup\limits_{v \in \mathbb{R}^d : \|v\| \neq 0} \|\mathbf{A}v\| / \|v\|$

**Definition 2** (Strong convexity)**.** Let $\beta > 0$. A function $f(\cdot)$ is $\beta$-strongly convex if and only if

$$f(y) \geq f(x) + \nabla^\top f(x)(y - x) + \frac{\beta}{2} \|x - y\|^2, \ \forall (x, y) \in \mathrm{dom}(f)$$

**Definition 3** (Lipschitz)**.** A function $f(\cdot)$ is $C$-Lipschitz if

$$\|f(x) - f(y)\| \leq C \|x - y\|, \ \forall (x, y) \in \mathrm{dom}(f).$$

In that case, $C$ is called the Lipschitz constant of $f$ and is denoted by $C \triangleq \mathrm{Lip}(f(x))$.

**Definition 4** (Smooth)**.** If $f(\cdot)$ is differentiable, then $f(\cdot)$ is $K$-smooth if

$$\|\nabla f(x) - \nabla f(y)\| \leq K \|x - y\|, \ \forall (x, y) \in \mathrm{dom}(f).$$

In that case, $K$ is called the gradient-Lipschitz constant of $f$ and is denoted by $K \triangleq \mathrm{Lip}_1(f(x))$.

**Definition 5** (Lipschitz-Hessian)**.** If $f(\cdot)$ is twice differentiable, then $f(\cdot)$ is $M$-Lipschitz Hessian if

$$\left\|\nabla^2 f(x) - \nabla^2 f(y)\right\|_{\mathrm{op}} \leq M \|x - y\|, \ \forall (x, y) \in \mathrm{dom}(f)$$

In that case, $M$ is called the Lipschitz-Hessian constant of $f$ and is denoted by $M \triangleq \mathrm{Lip}_2(f(x))$.

Throughout the manuscript, we will further make use of the next connections between Lipschitz coefficients and gradient bounds for differentiable functions.

**Corollary 5** (Nesterov (2013))**.** *Let $f(x)$ be a differentiable function $\forall x \in dom(f)$. Then, $f(x)$ is $C$-Lipschitz if and only if*

$$\|\nabla f(x)\| \leq C, \ \forall x \in \mathrm{dom}(f).$$

*If $f(x)$ is twice-differentiable $\forall x \in dom(f)$ then $f(x)$ is $K$-smooth if and only if*

$$\|\nabla^2 f(x)\|_{\mathrm{op}} \leq K, \ \forall x \in \mathrm{dom}(f).$$

# B    Example for Losses From an Exponential Family

We now present a few examples of commonly used loss functions in machine learning that can be viewed as the negative log-likelihood of an exponential-family model. A more comprehensive list can be found in Banerjee et al. (2005). Specifically, let

$$\ell\big(y, f(x; \theta)\big) \ = \ -\log\left(P\left(y|f(x; \theta)\right)\right),$$

where $P(y \mid f(x; \theta))$ belongs to a (discrete) exponential family. Throughout this paper, we adopt the following form of an exponential family:

$$\log\left(P\left(y|f(x; \theta)\right)\right) \ = \ f(x; \theta)^\top t(y) \ - \ \log\left(\sum_{\widetilde{y}=1}^{|\mathcal{Y}|} \exp\big\{ f(x; \theta)^\top t(\widetilde{y})\big\}\right) \ + \ \beta(y), \tag{14}$$

where $t(y)$ are the *natural statistics* and $f(x; \theta)$ are the *natural parameters*.[6] The term $\beta(y)$ depends only on $y$ (thus does not affect parameter learning) and ensures proper normalization. Below, we illustrate two popular examples of loss functions (see also (Martens, 2020, Sec. 9.2)) that arise naturally from this exponential-family framework.

---

[6]The above is a discrete version; for continuous $\mathcal{Y}$, one replaces the sum with an integral.

1. **Cross-Entropy Loss.** A standard approach in multi-class classification over $|\mathcal{Y}|$ classes is the softmax parameterization:

$$\log\left(P\left(y|f(x;\theta)\right)\right) = \left(f(x;\theta)\right)_y - \log\left(\sum_{\widetilde{y}=1}^{|\mathcal{Y}|} \exp\left\{\left(f(x;\theta)\right)_{\widetilde{y}}\right\}\right), \quad y, \widetilde{y} \in \{1, \ldots, |\mathcal{Y}|\}.$$

Here, $f(x;\theta)$ is a vector of length $|\mathcal{Y}|$. By defining $e_y$ as the one-hot vector with a 1 in the $y$-th entry and 0 elsewhere, we see that

$$\log P\left(y \mid f(x;\theta)\right) = f(x;\theta)^\top e_y - \log\left(\sum_{\widetilde{y}=1}^{|\mathcal{Y}|} \exp\left\{f(x;\theta)^\top e_{\widetilde{y}}\right\}\right),$$

thus matching (14) with natural statistics $t(y) = e_y$ and natural parameters $f(x;\theta)$. The corresponding loss,

$$\ell\left(y, f(x;\theta)\right) = -\log\left(P\left(y|f(x;\theta)\right)\right),$$

is the well-known cross-entropy.

2. **Mean Squared Error (MSE).** In a regression setting with a continuous target $y \in \mathbb{R}$, a unit-variance Gaussian model with mean $\mu = f(x;\theta)$ leads to

$$\log\left(P\left(y|f(x;\theta)\right)\right) = -\tfrac{1}{2}\left(y - f(x;\theta)\right)^2 = f(x;\theta)\,y - \frac{y^2}{2} - \frac{\left(f(x;\theta)\right)^2}{2}.$$

Comparing with (14), this corresponds to an exponential family whose natural statistics are $\left(y, y^2\right)$ and whose natural parameters are $\left(f(x;\theta), -\tfrac{1}{2}\right)$. The negative log-likelihood here,

$$\ell\left(y, f(x;\theta)\right) = -\log\left(P\left(y|f(x;\theta)\right)\right) = \tfrac{1}{2}\left(y - f(x;\theta)\right)^2 + \text{(constant)},$$

is precisely the mean squared error (MSE) loss up to an additive constant.

## B.1   Bregman Losses

Following (Banerjee et al., 2005, Thm. 4), whenever the representation $P(y|f(x;\theta))$ correspond to a regular exponential family, then the loss $-\log(P(y|f(x;\theta)))$ can be expressed as

$$-\log(P(y|f_\theta(x))) = d_\varphi(t(y), \mu(f_\theta(x))) + \log(b_\varphi(t(y))) + C$$

where $\mu(f_\theta(x)) = \mathbb{E}\left[t(y)\right]$ is the expected value of $t(y)$ using the underlying exponential family distribution, $d_\varphi(\cdot, \cdot)$ is a Bregman divergence and $C$ is a constant. As shown by (Banerjee et al., 2005, Table 1) (see also Das et al. (2025)), this result implies that many classical losses in machine learning, including cross-entropy and mean squared error, can be viewed as special cases of Bregman divergences, and further belong to the exponential family framework discussed in our work.

## B.2   Properties of the Cross-Entropy and MSE Losses

We now demonstrate how the assumptions on loss minimization, Hessian boundedness, and simplified second-order gradients follow for the two loss functions introduced above.

1. **Cross-Entropy Loss.** Recall the parameterization

$$\log\left(P\left(y|f(x;\theta)\right)\right) = \left(f(x;\theta)\right)_y - \log\left(\sum_{\widetilde{y}\in\mathcal{Y}} \exp\{\left(f(x;\theta)\right)_{\widetilde{y}}\}\right),$$

and let

$$\ell\big(y, f(x;\theta)\big) = -\log\left(P\left(y|f(x;\theta)\right)\right) = \log\left(\sum_{\widetilde{y}\in\mathcal{Y}}\exp\left\{(f(x;\theta))_{\widetilde{y}}\right\}\right) - (f(x;\theta))_y.$$

We focus first on the gradient of the *log-probability* itself, which is given by :

$$\nabla_f \log\left(P\left(y|f(x;\theta)\right)\right) = \nabla_f\left[(f(x;\theta))_y - \log\left(\sum_{\widetilde{y}\in\mathcal{Y}}\exp\left\{(f(x;\theta))_{\widetilde{y}}\right\}\right)\right]$$

$$= e_y - \text{softmax}\big(f(x;\theta)\big),$$

where $e_y$ is the one-hot vector selecting entry $y$, and $\text{softmax}\big(f(x;\theta)\big)$ is the vector of class probabilities assigned by the model.

**Zero Gradients Under Perfect Prediction.** For any training example $(x_i, y_i)$, if the model classifies it with perfect confidence, i.e.

$$\big(\text{softmax}(f(x_i; \hat{\theta}(\vec{1})))\big)_{y_i} = 1,$$

then $\nabla_f \log P\big(y_i \mid f(x_i; \hat{\theta}(\vec{1}))\big) = 0$. Consequently, if the model perfectly predicts *all* training labels, then all these gradients vanish simultaneously.

**Bounded Hessian.** Next, we show that the second derivative (the Hessian) of $\log\left(P\left(y|f(x;\theta)\right)\right)$ with respect to $f$ is bounded in norm. From the above,

$$\nabla_f \log\left(P\left(y|f(x;\theta)\right)\right) = e_y - \text{softmax}\big(f(x;\theta)\big),$$

so taking another derivative,

$$\nabla_f^2 \log\left(P\left(y|f(x;\theta)\right)\right) = -\nabla_f\big[\text{softmax}(f(x;\theta))\big].$$

Denote $\mathbf{C}_f \triangleq \nabla_f\big[\text{softmax}(f(x;\theta))\big]$. By the well-known derivative of softmax, the $(i,j)$th entry of $\mathbf{C}_f$ is

$$(\mathbf{C}_f)_{ij} = \frac{\partial}{\partial\big(f(x;\theta)\big)_j}\Big[\text{softmax}(f(x;\theta))_i\Big] = \text{softmax}(f(x;\theta))_i\big[\delta_{ij} - \text{softmax}(f(x;\theta))_j\big],$$

which implies:

$$(\mathbf{C}_f)_{ii} = \text{softmax}(f(x;\theta))_i \cdot \big[1 - \text{softmax}(f(x;\theta))_i\big],$$
$$(\mathbf{C}_f)_{ij} = -\text{softmax}(f(x;\theta))_i \cdot \text{softmax}(f(x;\theta))_j \quad (i \neq j).$$

Because each $\text{softmax}(f(x;\theta))_i \in [0,1]$, the entries of $\mathbf{C}_f$ lie in $[-1,1]$, and indeed one can show $\|\mathbf{C}_f\|$ is bounded by a constant (depending only on $|\mathcal{Y}|$, not on the dimension of the parameters). Hence $\nabla_f^2 \log\left(P\left(y|f(x;\theta)\right)\right) = -\mathbf{C}_f$ is also bounded in norm, establishing the desired Hessian bound.

2. **Mean Squared Error (MSE).** For the MSE loss arising from a unit-variance Gaussian,

$$\log\left(P\left(y|f(x;\theta)\right)\right) = -\tfrac{1}{2}\big[y - f(x;\theta)\big]^2,$$

the gradient with respect to $f(x;\theta)$ is simply

$$\nabla_f \log\left(P\left(y|f(x;\theta)\right)\right) = y - f(x;\theta).$$

Hence, if at $\hat{\theta}(\vec{1})$ the model predictions perfectly match all targets, this gradient becomes zero for each training pair, indicating perfect minimization of the training error.

**Bounded Hessian.** Since

$$\nabla_f^2 \log\left(P\left(y|f(x;\theta)\right)\right) \;=\; -\nabla_f^2 \left[\tfrac{1}{2}\left(y - f(x;\theta)\right)^2\right] \;=\; -\left(-\mathbb{I}_d\right) \;=\; \mathbb{I}_d,$$

the Hessian with respect to $f$ is simply the identity (for the one-dimensional $f$). Its norm is therefore trivially bounded by 1, and it does not depend on the dimension $d$ of the parameters in $\theta$. Moreover, the Hessian can be evaluated with no complicated operations—just the constant identity matrix at each sample.

## C  Gradient Bound for Minimizing Losses From the Exponential Family

Given a training set $\{(x_i, y_i)\}_{i=1}^n$ and the loss function (14) we derive gradient of the empirical risk (1) which we aim to minimize. To that end, we note that

$$
\begin{aligned}
n\nabla_\theta L(\mathscr{D}, \theta, \vec{1}) &= \nabla_\theta \left( \sum_{i=1}^n f^\top(x_i;\theta)t(y_i) - \log\left(\sum_{\tilde{y}\in\mathcal{Y}} \exp\{f^\top(x_i;\theta)t(\tilde{y})\}\right) + \beta(y_i)\right) \\
&= \sum_{i=1}^n \left(\nabla_\theta^\top f(x_i;\theta)t(y_i) - \frac{\sum_{\tilde{y}\in\mathcal{Y}} \nabla^\top f(x_i;\theta)t(\tilde{y})\exp\{f^\top(x_i;\theta)t(\tilde{y})\}}{\sum_{\tilde{y}_1\in\mathcal{Y}}\exp\{f^\top(x_i;\theta)t(\tilde{y}_1)\}}\right) \\
&= \sum_{i=1}^n \nabla_\theta^\top f(x_i;\theta)\left(t(y_i) - \frac{\sum_{\tilde{y}\in\mathcal{Y}} t(\tilde{y})\exp\{f^\top(x_i;\theta)t(\tilde{y})\}}{\sum_{\tilde{y}_1\in\mathcal{Y}}\exp\{f^\top(x_i;\theta)t(\tilde{y}_1)\}}\right) \\
&= \sum_{i=1}^n \nabla_\theta^\top f(x_i;\theta)(t(y_i) - \mathbb{E}_{\mathsf{y}\sim P_{\mathsf{y}|\mathsf{x}=x_i;\theta}}[t(\mathsf{y})])
\end{aligned}
$$

and the norm of this gradient is upper-bounded by

$$n\left\|\nabla_\theta L(\mathscr{D}, \theta, \vec{1})\right\| \leq \sum_{i=1}^n \|\nabla_\theta f(x_i;\theta)\| \left\|t(y_i) - \mathbb{E}_{\mathsf{y}\sim P_{\mathsf{y}|\mathsf{x}=x_i;\theta}}[t(\mathsf{y})]\right\|$$

Thus, whenever the features are Lipschitz, we have

$$\left\|\nabla_\theta L(\mathscr{D}, \theta, \vec{1})\right\| \leq \frac{C_f}{n}\sum_{i=1}^n \left\|t(y_i) - \mathbb{E}_{\mathsf{y}\sim P_{\mathsf{y}|\mathsf{x}=x_i;\theta}}[t(\mathsf{y})]\right\|$$

and we expect this upper bound to be small at the minimizer $\theta = \hat{\theta}(\vec{1})$. As a consequence, we note that the sum $\sum_{i=1}^n \|t(y_i) - \mathbb{E}_{\mathsf{y}\sim P_{\mathsf{y}|\mathsf{x}=x_i;\theta}}[t(\mathsf{y})]\|$ is in fact an upper bound on the gradient. Thus, when evaluated on $\theta = \hat{\theta}(\vec{1})$, we expect this quantity to be small, establishing the validity of the arguments from Section. 4.2, which claims that the additional factor that depends on $\bar{E}_n$ is indeed of negligible contribution to the final influence estimate.

## D  Proof of Closeness of $\hat{\theta}\left(\vec{1}\right)$ and $\hat{\theta}\left(\vec{1}^{n\setminus i}\right)$

We will use the next lemma throughout our proofs.

**Lemma 2.** *Let $\hat{\theta}\left(\vec{1}^{n\setminus i}\right)$ defined as in (1) and let $\frac{1}{n}\ell(z_i, \theta)$ be a differentiable function in $\theta$ for any $z_i \in \mathscr{D}$ and $\frac{1}{n}\ell(z_i, \theta) + \lambda\pi(\theta)$ be a $\mu$-strongly convex function in $\theta$ for any $z_i \in \mathscr{D}$. Then, $\forall i \in [n]$*

$$\left\|\hat{\theta}\left(\vec{1}^{n\setminus i}\right) - \hat{\theta}\left(\vec{1}\right)\right\| \leq \frac{2}{\mu}\cdot \max_{i\in[n]}\left\|\frac{1}{n}\nabla_\theta\ell\left(z_i, \hat{\theta}\left(\vec{1}\right)\right)\right\|$$

*Proof.* Similarly to the developments from (Wilson et al., 2020, App. B.1), we get that

$$\left\| \hat{\theta}\left(\vec{1}^{n\backslash i}\right) - \hat{\theta}\left(\vec{1}\right)\right\|^2 \le \frac{2}{\mu}\left|\left(\hat{\theta}\left(\vec{1}\right) - \hat{\theta}\left(\vec{1}^{n\backslash i}\right)\right)^\top \left(\nabla_\theta(L(\mathcal{D},\hat{\theta}\left(\vec{1}\right),\lambda,\vec{1}^{n\backslash i}) - L(\mathcal{D},\hat{\theta}\left(\vec{1}\right),\lambda,\vec{1})))\right|$$

$$\le \frac{2}{\mu n}\left|\left(\hat{\theta}\left(\vec{1}\right) - \hat{\theta}\left(\vec{1}^{n\backslash i}\right)\right)^\top \nabla_\theta \ell\left(z_i,\hat{\theta}\left(\vec{1}\right)\right)\right|$$

$$\le \frac{2}{\mu n}\left\|\hat{\theta}\left(\vec{1}\right) - \hat{\theta}\left(\vec{1}^{n\backslash i}\right)\right\| \left\|\nabla_\theta \ell\left(z_i,\hat{\theta}\left(\vec{1}\right)\right)\right\|$$

where all the steps are by Cauchy-Schwartz inequality. The proof follows by maximizing over $i$. $\qquad\square$

We note that whenever $\frac{1}{n}\ell(z_i,\theta)$ is Lipschitz, the upper bound is finite. Moreover, since we normalize by $n$, the bound will go to zero with $n$ whenever the gradient grows as $o(n)$, as is usually the case in many popular machine learning problems (see (Giordano et al., 2019b, Sec. 3)). We further note that under a more restrictive assumption that the $\ell(z_i,\theta)$ are Lipschitz then the bound is given by $\frac{2\tilde{C}}{\mu n}$ for $\tilde{C} = \max\limits_{i\in[n]}\left\|\nabla_\theta \ell\left(z_i,\hat{\theta}\left(\vec{1}\right)\right)\right\|$ and $\tilde{C} < \infty$.

## E   Proof of Lemma. 1

The proof uses the following lemma from Wilson et al. (2020):

**Lemma 3** (Optimizer Comparison, Wilson et al. (2020))**.** *Let*

$$x_{\varphi_1} \in \arg\min_x \varphi_1(x), \quad x_{\varphi_2} \in \arg\min_x \varphi_2(x).$$

*If each $\varphi_i$ is $\mu$-strongly convex and $\varphi_2 - \varphi_1$ is differentiable, then*

$$\frac{\mu}{2}\|x_{\varphi_1} - x_{\varphi_2}\|_2^2 \le \left|(x_{\varphi_1} - x_{\varphi_2})^\top \left(\nabla(\varphi_2 - \varphi_1)(x_{\varphi_1})\right)\right|.$$

*Proof.* For the sake of the proof, we will assume that the FIM and the Hessian are invertible matrices. Under the probabilistic interpretation of the loss elements, the overall loss function for $w^n = \vec{1}^{n\backslash i}$ is

$$L(\mathcal{D},\theta,\lambda,\vec{1}^{n\backslash i}) \triangleq \frac{1}{n}\sum_{j\ne i} -\log(P\left(y_j|f(x_j;\theta)\right)) + \lambda\pi(\theta)$$

and we assume that $P\left(y|f(x;\theta)\right)$ belongs to an exponential family whose natural parameters are the features $f(x;\theta)$, namely, $\log(P\left(y|f(x;\theta)\right)) = f^\top(x;\theta)t(y) - \log(\sum_{\tilde{y}=1}^{|\mathcal{Y}|}\exp\left\{f^\top(x;\theta)t(\tilde{y})\right\}) + \beta(y)$ for some natural statistics $t(y)$. For this model, we have

$$\nabla_\theta \log(P\left(y|f(x;\theta)\right)) = \nabla_\theta f(x;\theta)\nabla_f \log(P\left(y|f(x;\theta)\right)).$$

Thus, the approximated FIM, $\mathbf{F}(\mathcal{D},\theta)$, is given by

$$\mathbf{F}(\mathcal{D},\theta) = \frac{1}{n}\sum_{i=1}^n \mathbb{E}_{y\sim P_{y|x=x_i;\theta}}\left[\nabla_\theta f(x_i;\theta)\nabla_f \log(P\left(y|f(x_i;\theta)\right))\nabla_f^\top \log(P\left(y|f(x_i;\theta)\right))\nabla_\theta^\top f(x_i;\theta)\right]$$

$$= \frac{1}{n}\sum_{i=1}^n \nabla_\theta f(x_i;\theta)\mathbb{E}_{y\sim P_{y|x=x_i;\theta}}\left[-\nabla_f^2 \log(P\left(y|f(x_i;\theta)\right))\right]\nabla_\theta^\top f(x_i;\theta) \qquad (15a)$$

$$= -\frac{1}{n}\sum_{i=1}^n \nabla_\theta f(x_i;\theta)\nabla_f^2 \log(P\left(y_i|f(x_i;\theta)\right))\nabla_\theta^\top f(x_i;\theta)$$

where (15a) is by using classical properties of the exponential family, and where the last equality is since the Hessian of an exponential family with respect to the natural parameters $f$ is independent of $y$ (see App. G).

Moreover, we note that the Hessian of the loss is given by

$$
\begin{aligned}
\mathbf{H}(\theta, \vec{1}^{n\backslash i}) &= \nabla_\theta^2 L(\mathscr{D}, \theta, \vec{1}^{n\backslash i}) \\
&= \nabla_\theta^2 L(\mathscr{D}, \theta, \vec{1}^{n\backslash i} - \vec{1}) + \nabla_\theta^2 L(\mathscr{D}, \theta, \vec{1}) \\
&= \nabla_\theta^2 L(\mathscr{D}, \theta, \vec{1}^{n\backslash i} - \vec{1}) + \frac{1}{n} \sum_{i=1}^n \nabla_\theta^2 f(x_i; \theta) \nabla_f \log(P(y_i | f(x_i; \theta))) + \mathbf{F}(\mathscr{D}; \theta).
\end{aligned}
$$

We start by defining the next functions

$$
\begin{aligned}
\psi_1(\theta) &\triangleq 2L(\mathscr{D}, \theta, \lambda, \vec{1}^{n\backslash i}) = 2L(\mathscr{D}, \theta, \vec{1}^{n\backslash i}) + 2\lambda\pi(\theta), \\
\psi_2(\theta) &\triangleq -2b^\top(\hat{\theta}(\vec{1}), \vec{1}^{n\backslash i}) \cdot (\hat{\theta}(\vec{1}) - \theta) + (\hat{\theta}(\vec{1}) - \theta)^\top \nabla^2 L(\mathscr{D}, \hat{\theta}(\vec{1}), \vec{1}^{n\backslash i})(\hat{\theta}(\vec{1}) - \theta) + 2\lambda\pi(\theta), \\
\psi_3(\theta) &\triangleq -2b^\top(\hat{\theta}(\vec{1}), \vec{1}^{n\backslash i}) \cdot (\hat{\theta}(\vec{1}) - \theta) + (\hat{\theta}(\vec{1}) - \theta)^\top \cdot \mathbf{F} \cdot (\hat{\theta}(\vec{1}) - \theta) + 2\lambda\pi(\theta) \\
&= (\theta - (\hat{\theta}(\vec{1}) - \mathbf{F}^{-1} \cdot b(\hat{\theta}(\vec{1}), \vec{1}^{n\backslash i})))^\top \mathbf{F}(\theta - (\hat{\theta}(\vec{1}) - \mathbf{F}^{-1} \cdot b(\hat{\theta}(\vec{1}), \vec{1}^{n\backslash i}))) + 2\lambda\pi(\theta) + J
\end{aligned}
$$

where $J$ is a constant (which is independent of $\theta$) and $\mathbf{F}$ is an abbreviation for $\mathbf{F}(\mathscr{D}, \hat{\theta}(\vec{1}))$. We first note that the minimizer of $\psi_1$ is $\hat{\theta}(\vec{1}^{n\backslash i})$ and that the minimizer of $\psi_3$ is $\tilde{\theta}(\vec{1}^{n\backslash i})$ from (9).

We note that Assump. 1 and Assump. 2 guarantees that the overall loss, $L$, is $\mu$-strongly convex and that the difference $L(\mathscr{D}, \hat{\theta}(\vec{1}), \lambda, \vec{1}^{n\backslash i}) - L(\mathscr{D}, \hat{\theta}(\vec{1}), \lambda, \vec{1})$ is differentiable. Thus, using Lem. 2, which follows by applying the optimizer comparison lemma with $L(\mathscr{D}, \theta, \lambda, \vec{1}^{n\backslash i})$ and $L(\mathscr{D}, \theta, \lambda, \vec{1})$ allows us to derive the following upper bound

$$
\|\hat{\theta}(\vec{1}) - \hat{\theta}(\vec{1}^{n\backslash i})\| \leq \frac{2}{n\mu} \cdot \|\nabla_\theta \ell(z_i, \hat{\theta}(\vec{1}))\| \triangleq \frac{2g_i}{\mu}. \tag{16}
$$

The optimizer comparison lemma ([Wilson et al., 2020](), Lem. 1) with $\psi_1$ and $\psi_3$ and Cauchy-Schwartz inequality yields

$$
\begin{aligned}
\frac{\mu}{2}\|\hat{\theta}(\vec{1}^{n\backslash i}) - \tilde{\theta}(\vec{1}^{n\backslash i})\|^2 &\leq |(\hat{\theta}(\vec{1}^{n\backslash i}) - \tilde{\theta}(\vec{1}^{n\backslash i}))^\top (\nabla(\psi_3 - \psi_1)(\hat{\theta}(\vec{1}^{n\backslash i})))| \\
&\leq \|\hat{\theta}(\vec{1}^{n\backslash i}) - \tilde{\theta}(\vec{1}^{n\backslash i})\| \|(\nabla(\psi_3 - \psi_1)(\hat{\theta}(\vec{1}^{n\backslash i})))\|
\end{aligned}
$$

We divide both sides by $\|\hat{\theta}(\vec{1}^{n\backslash i}) - \tilde{\theta}(\vec{1}^{n\backslash i})\|$, and by using the triangle inequality we get

$$
\begin{aligned}
\frac{\mu}{2}\|\hat{\theta}(\vec{1}^{n\backslash i}) - \tilde{\theta}(\vec{1}^{n\backslash i})\| &\leq \|\nabla(\psi_3 - \psi_1)(\hat{\theta}(\vec{1}^{n\backslash i}))\| \\
&\leq \|\nabla(\psi_3 - \psi_2)(\hat{\theta}(\vec{1}^{n\backslash i})) + \nabla(\psi_2 - \psi_1)(\hat{\theta}(\vec{1}^{n\backslash i}))\| \\
&\leq \|\nabla(\psi_3 - \psi_2)(\hat{\theta}(\vec{1}^{n\backslash i}))\| + \|\nabla(\psi_2 - \psi_1)(\hat{\theta}(\vec{1}^{n\backslash i}))\| \\
&\leq \|\nabla^2 L(\mathscr{D}, \hat{\theta}(\vec{1}), \vec{1}^{n\backslash i}) - \mathbf{F}(\mathscr{D}, \hat{\theta}(\vec{1}))\| \|\hat{\theta}(\vec{1}) - \hat{\theta}(\vec{1}^{n\backslash i})\| + \|(\nabla(\psi_2 - \psi_1)(\hat{\theta}(\vec{1}^{n\backslash i})))\| \\
&= \|\nabla^2 L(\mathscr{D}, \hat{\theta}(\vec{1}), \vec{1}^{n\backslash i} - \vec{1}) + \frac{1}{n} \sum_{i=1}^n \nabla_\theta^2 f(x_i; \hat{\theta}(\vec{1})) \nabla_f \log(P(y_i | f(x_i; \hat{\theta}(\vec{1}))))\| \|\hat{\theta}(\vec{1}) - \hat{\theta}(\vec{1}^{n\backslash i})\| \\
&\qquad + \|(\nabla(\psi_2 - \psi_1)(\hat{\theta}(\vec{1}^{n\backslash i})))\| \\
&\leq \frac{g_i}{n\mu} \cdot \| - \nabla_\theta f(x_i; \hat{\theta}(\vec{1})) \nabla_f^2 \log(P(y_i | f(x_i; \hat{\theta}(\vec{1})))) \nabla_\theta^\top f(x_i; \hat{\theta}(\vec{1})) \\
&\qquad + \sum_{i=1}^n \nabla_\theta^2 f(x_i; \hat{\theta}(\vec{1})) \nabla_f \log(P(y_i | f(x_i; \hat{\theta}(\vec{1}))))\| + \frac{M g_i^2}{2\mu^2}
\end{aligned} \tag{17} \tag{18}
$$

where (17) is since the differences $\psi_3 - \psi_2$ and $\psi_2 - \psi_1$ are differentiable and where (18) is by using the next bound:

$$\|(\nabla(\psi_2 - \psi_1)(\hat{\theta}(\vec{1}^{n\backslash i})))\|$$
$$= 2\|b(\hat{\theta}(\vec{1}), \vec{1}^{n\backslash i}) + \nabla^2 L(\mathscr{D}, \hat{\theta}(\vec{1}), \vec{1}^{n\backslash i})(\hat{\theta}(\vec{1}^{n\backslash i}) - \hat{\theta}(\vec{1})) - \nabla L(\mathscr{D}, \hat{\theta}(\vec{1}^{n\backslash i}), \vec{1}^{n\backslash i})\|$$
$$= 2\|\nabla L(\mathscr{D}, \hat{\theta}(\vec{1}), \vec{1}^{n\backslash i}) + \nabla^2 L(\mathscr{D}, \hat{\theta}(\vec{1}), \vec{1}^{n\backslash i})(\hat{\theta}(\vec{1}^{n\backslash i}) - \hat{\theta}(\vec{1})) - \nabla L(\mathscr{D}, \hat{\theta}(\vec{1}^{n\backslash i}), \vec{1}^{n\backslash i})\| \tag{19a}$$
$$\leq M \cdot \left\|\hat{\theta}(\vec{1}^{n\backslash i}) - \hat{\theta}(\vec{1})\right\|^2 \tag{19b}$$
$$\leq \frac{4Mg_i^2}{\mu^2}$$

where (19a) is by the structure and the convexity and differentiability assumptions on $L$, leading to $\nabla L(\mathscr{D}, \hat{\theta}(\vec{1}), \vec{1}) = 0$, (19b) implied by the Hessian Lipschitzness of $L$ (see also [3, Lem. 1.2.4]) and the last inequality is by Lem. 2.

We further use the triangle inequality to get the next upper bound

$$\frac{\mu}{2}\|\hat{\theta}(\vec{1}^{n\backslash i}) - \tilde{\theta}(\vec{1}^{n\backslash i})\| \leq \frac{g_i}{n\mu}(\|\nabla_\theta f(x_i; \hat{\theta}(\vec{1}))\nabla_f^2 \log(P(y_i|f(x_i; \hat{\theta}(\vec{1}))))\nabla_\theta^\top f(x_i; \hat{\theta}(\vec{1}))\|$$
$$+ \sum_{i=1}^n \|\nabla_\theta^2 f(x_i; \hat{\theta}(\vec{1}))\nabla_f \log(P(y_i|f(x_i; \hat{\theta}(\vec{1}))))\|) + \frac{Mg_i^2}{2\mu^2}$$

and by using Assumption. 5, Assumption. 7 and the boundedness of the Hessian of the loss relative to the features (see App. B.2) we get the final bound

$$\|\hat{\theta}(\vec{1}^{n\backslash i}) - \tilde{\theta}(\vec{1}^{n\backslash i})\| \leq \frac{2Qg_i}{n\mu^2}\|\nabla_\theta f(x_i; \hat{\theta}(\vec{1}))\|^2 + \frac{Mg_i^2}{\mu^3}$$
$$+ \frac{2g_i}{n\mu^2}\sum_{i=1}^n \|\nabla_\theta^2 f(x_i; \hat{\theta}(\vec{1}))\nabla_f \log(P(y_i|f(x_i; \hat{\theta}(\vec{1}))))\|$$
$$\leq \frac{2QC_f^2 g_i}{n\mu^2} + \frac{Mg_i^2}{\mu^3} + \frac{2g_i\tilde{C}_f}{n\mu^2}\sum_{i=1}^n \|\nabla_f \log(P(y_i|f(x_i; \hat{\theta}(\vec{1}))))\|$$

where $Q$ is a constant s.t. $\left\|\nabla_f^2 \log(P(y|f(x; \theta)))\right\| \leq Q$. $\qquad\square$

We now emphasize how the third term disappears whenever our model interpolates the training data (namely, $\ell(z_i, \hat{\theta}(\vec{1})) = 0, \forall i \in [n]$). In that case, we have $P(y_i|f(x_i; \hat{\theta}(\vec{1}))) = 1, \forall i \in [n]$ [7]. Thus, following the notation of App. G we have that $\mathbb{E}_{\mathsf{y}\sim P_{\mathsf{y}|\mathsf{x}=x_i;\hat{\theta}(\vec{1})}}[t(\mathsf{y})] = t(y_i)$ and since $\nabla_f \log(P(y_i|f(x_i; \hat{\theta}(\vec{1})))) = t(y_i) - \mathbb{E}_{\mathsf{y}\sim P_{\mathsf{y}|\mathsf{x}=x_i;\hat{\theta}(\vec{1})}}[t(\mathsf{y})]$ we get that the third term is zero.

## F  Comment on Lemma. 1 When $\pi(\theta)$ is Twice-Differentiable

Whenever $\pi(\theta)$ is twice differentiable, an equivalent argument to that of Lemma. 1 can be stated without the usage of a proximal operator. Specifically, since in this case the entire loss elements $\frac{1}{n}\ell(z_i, \theta) + \lambda\pi(\theta)$ can be approximated using a second-order Taylor expansion, and a solution that uses $\mathbf{C}\left(\hat{\theta}(\vec{1}), \vec{1}\right) = \mathbf{F}\left(\mathscr{D}, \hat{\theta}(\vec{1})\right) + \lambda\nabla^2\pi\left(\hat{\theta}(\vec{1})\right)$ leads to similar arguments as those from App. E. For this approximation, we define the solution via

$$\tilde{\theta}\left(\vec{1}^{n\backslash i}\right) \triangleq \hat{\theta}(\vec{1}) - \left(\mathbf{F}\left(\mathscr{D}, \hat{\theta}(\vec{1})\right) + \lambda\nabla^2\pi\left(\hat{\theta}(\vec{1})\right)\right)^{-1} b\left(\hat{\theta}(\vec{1}), \vec{1}^{n\backslash i}\right)$$

and a similar analysis to that of App. E can be carried out and to lead to similar guarantees. An example for such arguments from a similar application can be found in (Wilson et al., 2020, Thm. 2).

---

[7]In the continuous case, this amounts to $P(y_i|f(x_i; \hat{\theta}(\vec{1})))$ converging to a delta-function, concentrated around the value $y_i$

## G  Fisher Information Matrix for Exponential Families

Using the fact that the distribution $P(y|f(x;\theta))$ belongs to an exponential family, namely

$$\log(P(y|f(x;\theta))) = f^\top(x;\theta)t(y) - \log\left(\sum_{\tilde{y}=1}^{|\mathcal{Y}|}\exp\left\{f^\top(x;\theta)t(\tilde{y})\right\}\right) + \beta(y),$$

we can directly evaluate the terms $\mathbb{E}_{\mathsf{y}\sim P_{\mathsf{y}|\mathsf{x}=x_i;\theta}}\left[\nabla_f\log(P(\mathsf{y}|f(x;\theta)))\nabla_f^\top\log(P(\mathsf{y}|f(x;\theta)))\right]$ and
$\mathbb{E}_{\mathsf{y}\sim P_{\mathsf{y}|\mathsf{x}=x_i;\theta}}\left[-\nabla_f^2\log(P(\mathsf{y}|f(x_i;\theta)))\right]$ to establish the desired equality. First, we find that:

$$\nabla_f\log(P(y|f(x;\theta))) = \nabla_f\left(f^\top(x;\theta)t(y) - \log\left(\sum_{y\in\mathcal{Y}}\exp\left\{f^\top(x;\theta)t(y)\right\}\right)\right)$$
$$= t(y) - \mathbb{E}_{\mathsf{y}\sim P_{\mathsf{y}|\mathsf{x}=x_i;\theta}}[t(\mathsf{y})]$$

and

$$\mathbb{E}_{\mathsf{y}\sim P_{\mathsf{y}|\mathsf{x}=x_i;\theta}}\left[\nabla_f\log(P(\mathsf{y}|f(x;\theta)))\nabla_f^\top\log(P(\mathsf{y}|f(x;\theta)))\right]$$
$$= \mathbb{E}_{\mathsf{y}\sim P_{\mathsf{y}|\mathsf{x}=x_i;\theta}}\left[(t(\mathsf{y}) - \mathbb{E}_{\mathsf{y}\sim P_{\mathsf{y}|\mathsf{x}=x_i;\theta}}[t(\mathsf{y})])(t(\mathsf{y}) - \mathbb{E}_{\mathsf{y}\sim P_{\mathsf{y}|\mathsf{x}=x_i;\theta}}[t(\mathsf{y})])^\top\right].$$

Next, we observe that:

$$-\nabla_f^2\log(P(y|f(x;\theta))) = \nabla_f\left(\frac{\sum_{\tilde{y}\in\mathcal{Y}}t(\tilde{y})\exp\left\{f^\top(x;\theta)t(\tilde{y})\right\}}{\sum_{\tilde{y}_1\in\mathcal{Y}}\exp\left\{f^\top(x;\theta)t(\tilde{y}_1)\right\}}\right)$$
$$= \mathbb{E}_{\mathsf{y}\sim P_{\mathsf{y}|\mathsf{x}=x_i;\theta}}\left[t(\mathsf{y})t^\top(\mathsf{y})\right] - (\mathbb{E}_{\mathsf{y}\sim P_{\mathsf{y}|\mathsf{x}=x_i;\theta}}[t(\mathsf{y})])(\mathbb{E}_{\mathsf{y}\sim P_{\mathsf{y}|\mathsf{x}=x_i;\theta}}[t(\mathsf{y})])^\top$$
$$= \mathbb{E}_{\mathsf{y}\sim P_{\mathsf{y}|\mathsf{x}=x_i;\theta}}\left[(t(\mathsf{y}) - \mathbb{E}_{\mathsf{y}\sim P_{\mathsf{y}|\mathsf{x}=x_i;\theta}}[t(\mathsf{y})])(t(\mathsf{y}) - \mathbb{E}_{\mathsf{y}\sim P_{\mathsf{y}|\mathsf{x}=x_i;\theta}}[t(\mathsf{y})])^\top\right].$$

Moreover, we note that this final result holds for any $y$. This concludes the proof. $\square$

## H  Proof of Theorem. 1

*Proof.* We start by writing the Taylor expansion of $T(\tilde{\theta}(\vec{1}^{n\backslash i}),\vec{1}^{n\backslash i})$ around $\hat{\theta}(\vec{1}^{n\backslash i})$ to get [8]:

$$T(\tilde{\theta}(\vec{1}^{n\backslash i}),\vec{1}^{n\backslash i}) = T(\hat{\theta}(\vec{1}^{n\backslash i}),\vec{1}^{n\backslash i}) + \nabla_\theta^\top T(\hat{\theta}(\vec{1}^{n\backslash i}),\vec{1}^{n\backslash i})(\tilde{\theta}(\vec{1}^{n\backslash i}) - \hat{\theta}(\vec{1}^{n\backslash i})) \qquad (20)$$
$$+ \frac{1}{2}(\tilde{\theta}(\vec{1}^{n\backslash i}) - \hat{\theta}(\vec{1}^{n\backslash i}))^\top\nabla_\theta^2 T(\theta_{\mathrm{mid}}(\vec{1}^{n\backslash i}),\vec{1}^{n\backslash i})(\tilde{\theta}(\vec{1}^{n\backslash i}) - \hat{\theta}(\vec{1}^{n\backslash i}))$$

where $\theta_{\mathrm{mid}}(\vec{1}^{n\backslash i}) = \hat{\theta}(\vec{1}^{n\backslash i}) + \kappa\cdot(\tilde{\theta}(\vec{1}^{n\backslash i}) - \hat{\theta}(\vec{1}^{n\backslash i}))$ for some $\kappa\in[0,1]$. By (20) and by the Lipschitz assumptions on $T$ we get

$$\|T(\tilde{\theta}(\vec{1}^{n\backslash i}),\vec{1}^{n\backslash i}) - T(\hat{\theta}(\vec{1}^{n\backslash i}),\vec{1}^{n\backslash i})\|$$
$$= \|\nabla_\theta^\top T(\hat{\theta}(\vec{1}^{n\backslash i}),\vec{1}^{n\backslash i})(\tilde{\theta}(\vec{1}^{n\backslash i}) - \hat{\theta}(\vec{1}^{n\backslash i}))$$
$$+ \frac{1}{2}(\tilde{\theta}(\vec{1}^{n\backslash i}) - \hat{\theta}(\vec{1}^{n\backslash i}))^\top\nabla_\theta^2 T(\theta_{\mathrm{mid}}(\vec{1}^{n\backslash i}),\vec{1}^{n\backslash i})(\tilde{\theta}(\vec{1}^{n\backslash i}) - \hat{\theta}(\vec{1}^{n\backslash i}))\|$$
$$\leq \|\nabla_\theta T(\hat{\theta}(\vec{1}^{n\backslash i}),\vec{1}^{n\backslash i})\|\|\tilde{\theta}(\vec{1}^{n\backslash i}) - \hat{\theta}(\vec{1}^{n\backslash i})\| \qquad (21a)$$
$$+ \frac{1}{2}\|\nabla_\theta^2 T(\theta_{\mathrm{mid}}(\vec{1}^{n\backslash i}),\vec{1}^{n\backslash i})\|_{\mathrm{op}}\|\tilde{\theta}(\vec{1}^{n\backslash i}) - \hat{\theta}(\vec{1}^{n\backslash i})\|^2$$
$$\leq C_{T_1}\|\tilde{\theta}(\vec{1}^{n\backslash i}) - \hat{\theta}(\vec{1}^{n\backslash i})\| + \frac{1}{2}C_{T_2}\|\tilde{\theta}(\vec{1}^{n\backslash i}) - \hat{\theta}(\vec{1}^{n\backslash i})\|^2. \qquad (21b)$$

---

[8]the existence of the Taylor expansion of $T$ is guaranteed by Assumption. 6

The proof is completed by substituting (10) into (21b). To prove (12), we write the expansion of $T(\hat{\theta}(\vec{1}^{n\backslash i}))$ around $\hat{\theta}(\vec{1})$, to get

$$
\|T(\hat{\theta}(\vec{1}^{n\backslash i}), \vec{1}^{n\backslash i}) - T(\hat{\theta}(\vec{1}), \vec{1}^{n\backslash i}) - \nabla_\theta T(\hat{\theta}(\vec{1}), \vec{1}^{n\backslash i})(\tilde{\theta}(\vec{1}^{n\backslash i}) - \hat{\theta}(\vec{1}))\|
$$
$$
= \|\nabla_\theta T(\hat{\theta}(\vec{1}), \vec{1}^{n\backslash i})(\tilde{\theta}(\vec{1}^{n\backslash i}) - \hat{\theta}(\vec{1}^{n\backslash i}))
$$
$$
+ \frac{1}{2}(\hat{\theta}(\vec{1}^{n\backslash i}) - \hat{\theta}(\vec{1}))^\top \nabla_\theta^2 T(\tilde{\theta}_{\mathrm{mid}}, \vec{1}^{n\backslash i})(\hat{\theta}(\vec{1}^{n\backslash i}) - \hat{\theta}(\vec{1}))\|
$$
$$
\leq C_{T_1}\|\tilde{\theta}(\vec{1}^{n\backslash i}) - \hat{\theta}(\vec{1}^{n\backslash i})\| + \frac{1}{2}C_{T_2}\|\hat{\theta}(\vec{1}) - \hat{\theta}(\vec{1}^{n\backslash i})\|^2
$$

where $\tilde{\theta}_{\mathrm{mid}} = \hat{\theta}(\vec{1}^{n\backslash i}) + \kappa \cdot (\hat{\theta}(\vec{1}) - \hat{\theta}(\vec{1}^{n\backslash i}))$ for some $\kappa \in [0, 1]$. Substituting (10) and (16) concludes the proof. $\square$

## I Proofs of Corollary. 1 - Corollary. 4

### I.1 Proof of Corollary. 1

We now show how to use Theorem. 1 to approximate LOOCV with similar guarantees to the Hessian-based technique from (Wilson et al., 2020). Throughout the proof, we will use a refined version of (21b), which requires the Lipschitzness of the $T(\cdot, \vec{1}^{n\backslash i})$ only at $\hat{\theta}(\vec{1})$. We start by defining $\mathrm{ACV} \triangleq \frac{1}{n}\sum_{i=1}^n \ell(z_i, \tilde{\theta}(\vec{1}^{n\backslash i}))$ and recall that $\mathrm{CV} \triangleq \frac{1}{n}\sum_{i=1}^n \ell(z_i, \hat{\theta}(\vec{1}^{n\backslash i}))$. Then, similarly to App. H we get

$$
|\mathrm{ACV} - \mathrm{CV}|
$$
$$
= \left|\frac{1}{n}\sum_{i=1}^n \ell(z_i, \tilde{\theta}(\vec{1}^{n\backslash i})) - \ell(z_i, \hat{\theta}(\vec{1}^{n\backslash i}))\right|
$$
$$
\leq \frac{1}{n}\sum_{i=1}^n \left|\ell(z_i, \tilde{\theta}(\vec{1}^{n\backslash i})) - \ell(z_i, \hat{\theta}(\vec{1}^{n\backslash i}))\right|
$$
$$
\leq \frac{1}{n}\sum_{i=1}^n \left\|\nabla_\theta \ell(z_i, \hat{\theta}(\vec{1}^{n\backslash i}))\right\| \left(\frac{2QC_f^2 \tilde{g}_i}{n^2\mu^2} + \frac{M\tilde{g}_i^2}{n^2\mu^3} + \frac{2\tilde{g}_i \tilde{C}_f \bar{E}_n}{n\mu^2}\right) \tag{22a}
$$
$$
+ \frac{1}{2}\mathrm{Lip}(\nabla_\theta \ell(z_i, \theta))\left(\frac{2QC_f^2 \tilde{g}_i}{n^2\mu^2} + \frac{M\tilde{g}_i^2}{n^2\mu^3} + \frac{2\tilde{g}_i \tilde{C}_f \bar{E}_n}{n\mu^2}\right)^2
$$
$$
\leq \frac{1}{n}\sum_{i=1}^n \left(\left\|\nabla_\theta \ell(z_i, \hat{\theta}(\vec{1}))\right\| + \mathrm{Lip}(\nabla_\theta \ell(z_i, \theta))\left(\frac{4\tilde{g}_i^2}{n^2\mu^2}\right)\right)\left(\frac{2QC_f^2 \tilde{g}_i}{n^2\mu^2} + \frac{M\tilde{g}_i^2}{n^2\mu^3} + \frac{2\tilde{g}_i \tilde{C}_f \bar{E}_n}{n\mu^2}\right) \tag{22b}
$$
$$
+ \frac{1}{2}\mathrm{Lip}(\nabla_\theta \ell(z_i, \theta))\left(\frac{2QC_f^2 \tilde{g}_i}{n^2\mu^2} + \frac{M\tilde{g}_i^2}{n^2\mu^3} + \frac{2\tilde{g}_i \tilde{C}_f \bar{E}_n}{n\mu^2}\right)^2
$$

where (22a) is by using (21a) together with the bound from Theorem. 1 and by replacing the Lipschitz constants $C_{T_1}$ and $C_{T_2}$ of the objective with the corresponding gradients from (21a) and (22b) is by using the Taylor expansion of $\nabla_\theta \ell(z_i, \hat{\theta}(\vec{1}^{n\backslash i}))$ around $\hat{\theta}(\vec{1})$ and by using Lemma. 2. Expanding this expression

yields

$$
\begin{aligned}
|\text{ACV} - \text{CV}| \leq & \left( \frac{2QC_f^2}{\mu^2 n^2} + \frac{2\tilde{C}_f \bar{E}_n}{\mu^2 n} \right) \cdot \frac{1}{n} \sum_{i=1}^n \left\| \nabla_\theta \ell(z_i, \hat{\theta}(\vec{1})) \right\|^2 + \left( \frac{M}{\mu^3 n^2} \right) \cdot \frac{1}{n} \sum_{i=1}^n \left\| \nabla_\theta \ell(z_i, \hat{\theta}(\vec{1})) \right\|^3 \\
& + \left( \frac{8QC_f^2}{\mu^4 n^4} + \frac{8\tilde{C}_f \bar{E}_n}{\mu^4 n^3} \right) \cdot \frac{1}{n} \sum_{i=1}^n \text{Lip}(\nabla_\theta \ell(z_i, \theta)) \left\| \nabla_\theta \ell(z_i, \hat{\theta}(\vec{1})) \right\|^3 \\
& + \left( \frac{4M}{\mu^5 n^4} \right) \cdot \frac{1}{n} \sum_{i=1}^n \text{Lip}(\nabla_\theta \ell(z_i, \theta)) \left\| \nabla_\theta \ell(z_i, \hat{\theta}(\vec{1})) \right\|^4 \\
& + \left( \frac{2Q^2 C_f^4}{\mu^4 n^4} + \frac{2\tilde{C}_f^2 \bar{E}_n^2}{\mu^4 n^2} \right) \cdot \frac{1}{n} \sum_{i=1}^n \text{Lip}(\nabla_\theta \ell(z_i, \theta)) \left\| \nabla_\theta \ell(z_i, \hat{\theta}(\vec{1})) \right\|^2 \\
& + \left( \frac{2QC_f^2 M}{n^4 \mu^5} \right) \cdot \frac{1}{n} \sum_{i=1}^n \text{Lip}(\nabla_\theta \ell(z_i, \theta)) \left\| \nabla_\theta \ell(z_i, \hat{\theta}(\vec{1})) \right\|^3 \\
& + \left( \frac{M^2}{n^4 \mu^6} \right) \cdot \frac{1}{n} \sum_{i=1}^n \text{Lip}(\nabla_\theta \ell(z_i, \theta)) \left\| \nabla_\theta \ell(z_i, \hat{\theta}(\vec{1})) \right\|^4 \\
& + \left( \frac{M\tilde{C}_f \bar{E}_n}{n^3 \mu^5} \right) \cdot \frac{1}{n} \sum_{i=1}^n \text{Lip}(\nabla_\theta \ell(z_i, \theta)) \left\| \nabla_\theta \ell(z_i, \hat{\theta}(\vec{1})) \right\|^3 \\
& + \left( \frac{2Q\tilde{C}_f C_f^2 \bar{E}_n}{n^3 \mu^4} \right) \cdot \frac{1}{n} \sum_{i=1}^n \text{Lip}(\nabla_\theta \ell(z_i, \theta)) \left\| \nabla_\theta \ell(z_i, \hat{\theta}(\vec{1})) \right\|^2
\end{aligned}
$$

whose decay rate is dictated by the first two terms and is given by $O\left( \frac{C_f^2 B_{02}}{\mu^2 n^2} + \frac{\tilde{C}_f \bar{E}_n B_{02}}{\mu^2 n} + \frac{M B_{03}}{\mu^3 n^2} \right)$. $\qquad \square$

### I.2 Proof of Corollary. 2

The proof follows similarly to that from (Suriyakumar and Wilson, 2022) by using the bound $\tilde{g}_i \leq G$ in (10) and then using the Gaussian mechanism for differential privacy (Dwork et al., 2014, App. A). $\qquad \square$

We note that Corollary. 2 parallels a similar result to that of Proposition. 2, with different Lipschitz constants and with an additional term that depends on $\bar{E}_n$.

### I.3 Proof of Corollary. 3

The proof is by substituting $\tilde{g}_i = \|\nabla_\theta \ell(z_i, \hat{\theta}(\vec{1}))\|$ in (11) and (12) and maximizing over $i$. $\qquad \square$

We note that this proof parallels a similar result to that of Proposition. 3, with two additional terms: one that depends on $\bar{E}_n$ and the other that depends on the Lipschitz coefficient of the features $C_f$.

### I.4 Proof of Corollary. 4

By using the definition of $T$ from (2) and using the linearity of expectation and the triangle inequality we get that the Lipschitz coefficient of $T$ from (2), $C_{T_1}$, is given by $2C_f$. Then, the proof follows by substituting $\tilde{g}_i = \|\nabla_\theta \ell(z_i, \hat{\theta}(\vec{1}))\|$ in (11) and maximizing over $i$. $\qquad \square$

## J Experimental Details

All experiments were implemented using the PyTorch (Paszke et al., 2019) framework and ran on an NVIDIA A100 GPU.

### J.1 Datasets

#### J.1.1 Logistic Regression With Non-Linear Natural Parameter Map

To generate the data, we first draw a fixed Random Fourier Feature (RFF) map $\phi : \mathbb{R}^d \to \mathbb{R}^p$ by sampling $W \in \mathbb{R}^{p \times d}$ with i.i.d. Gaussian entries and $b \in \mathbb{R}^p$ with i.i.d. uniform entries on $[0, 2\pi]$, and setting

$$\phi(x) = \sqrt{\frac{2}{p}} \cos(Wx + b).$$

We then sample covariates $x_i \sim \mathcal{N}(0, \mathbb{I}_d)$ i.i.d., and generate labels $y_i \in \{0, 1\}$ conditionally independently according to a Bernoulli model with logistic link,

$$y_i \mid x_i \sim \text{Bernoulli}\left(\sigma(\eta_i^\star)\right), \qquad \eta_i^\star = \theta^{\star\top}\phi(x_i) + \frac{\alpha}{2}\|\theta^\star\|_2^2,$$

where $\sigma(t) = 1/(1 + e^{-t})$ denotes the sigmoid function. The parameter $\theta^\star$ was sampled as a vector uniformly distributed on the unit sphere. Finally, the RFF parameters $(W, b)$ are held fixed after sampling, and both training and test sets are drawn independently from the same generative process.

#### J.1.2 Friedman–1

We use the Friedman–1 regression benchmark Friedman (1991). We draw feature vectors with coordinates i.i.d. $\text{Unif}(-1, 1)$ and generate responses as

$$y_n = 10 \sin\left(\pi x_n^{(1)} x_n^{(2)}\right) + 20\left(x_n^{(3)} - \tfrac{1}{2}\right)^2 + 10 x_n^{(4)} + 5 x_n^{(5)} + 0.1 z_n, \qquad z_n \sim \mathcal{N}(0, 1).$$

Only the first five coordinates are informative; the remaining $d - 5$ are nuisance. To study a higher-dimensional, sparse regime, we append 300 i.i.d. random features independent of $(\boldsymbol{x}, y)$ and train with $\ell_1$ regularization. Unless stated otherwise, we use $N = 2000$ samples and $d = 100$ base features (before appending the 300 random features).

### J.2 Models

All models were trained either using a cross-entropy loss or using an MSE loss, implemented via `torch.nn.CrossEntropyLoss()` and `torch.nn.MSELoss()`. We have conducted our experiments with an $L_1$ regularization, namely, $\pi(\theta) = \|\theta\|_1^2$.

**Logistic Regression With Non-Linear Natural Parameter Map:** Recall that our model is characterized by the conditional distribution

$$p(y \mid \eta) = \sigma(\eta)^y \left(1 - \sigma(\eta)\right)^{1-y} = \exp\left\{y\eta - A(\eta)\right\}, \qquad A(\eta) = \log\left(1 + e^\eta\right),$$

which is a regular one-parameter exponential family with natural parameter $\eta$ and sufficient statistic $T(y) = y$. In our setting, the natural parameter is given by $\eta = f(x; \theta) = \theta^\top \phi(x) + \frac{\alpha}{2}\|\theta\|_2^2$, where $\phi(\cdot)$ is fixed. The log-partition function $A$ is twice continuously differentiable and satisfies

$$A''(\eta) = \sigma(\eta)\left(1 - \sigma(\eta)\right) \leq \tfrac{1}{4} \quad \text{for all } \eta \in \mathbb{R},$$

so the curvature of the log-likelihood with respect to the natural parameter is uniformly bounded. Consequently, the per-sample loss $\ell(z_i, \theta) = -\log p(y_i \mid f(x_i; \theta))$ is smooth in $\theta$ (since $f$ is smooth) and its second-order behavior is controlled via the chain rule by the bounded curvature of $A$ together with bounds on the derivatives of $f$. Finally, adding the Ridge term $\frac{\lambda}{2}\|\theta\|_2^2$ ensures strong convexity of the objective (in particular, for $\lambda > 0$). However, we note that the function $f(x; \theta)$ is non-linear in $\theta$, as desired.

**Two-Layer Network:** We have tested our models with a two-layer fully-connected network, with continuous output (namely, no activation on the last layer). The activation function for the hidden layer was chosen as `SeLU` activation. The other parameters we have used were:

1. The width of the hidden layer was chosen to be 100.

2. We trained the model using the SGD optimizer, with a learning rate of $10^{-4}$, batch size of 32, and a weight-decay of $10^{-4}$.

3. We also added an $\ell_1$ regularization component of strength $10^{-2}$.

4. We varied the number of epochs from 1 to 10.

## J.3 Details for Logistic Regression With Non-Linear Natural Parameter Map

For each value of the regularization parameter $\lambda$, we train the model by minimizing (13) using the Adam optimizer with batches of size 4096, 15 epochs, and a learning rate of $2 \cdot 10^{-2}$. Influence computations are performed at $\widehat{\theta}$ using $K$-fold splits (with $K = 10$): for each fold, we form the mean validation loss

$$L_{\mathrm{val}}(\theta) = \frac{1}{|\mathscr{V}|} \sum_{i \in \mathscr{V}} \ell(z_i, \theta),$$

and compute its gradient $g_{\mathrm{val}} = \nabla_\theta L_{\mathrm{val}}(\widehat{\theta})$ via automatic differentiation. We then solve a linear system $Au = v$ where $A$ is either (i) the observed Hessian $\mathbf{H} = \nabla_\theta^2 L_{\mathrm{train}}(\widehat{\theta}) + \lambda \mathbb{I}_d$ or (ii) the Fisher information matrix $\mathbf{F}$ evaluated at $\widehat{\theta}$. Both methods use the *same* linear solver and differ only in the matrix–vector product: Hessian-vector products are computed as $u \mapsto \nabla_\theta^2 L_{\mathrm{train}}(\widehat{\theta})\, u$ using forward-over-reverse automatic differentiation, while Fisher-vector products are computed as $u \mapsto J^\top W J u$ using Jacobian-vector products (JVPs) and vector-Jacobian products (VJPs), with $W = \mathrm{diag}(p_i(1 - p_i))/|\mathscr{T}|$ and $p_i = \sigma(\eta_i(\widehat{\theta}))$. The linear systems are solved with conjugate gradient up to tolerance $10^{-6}$ and a maximum of 300 iterations. Finally, the influence-based estimate of the fold CV loss is obtained by applying the corresponding first-order parameter update to $\widehat{\theta}$ (i.e., the approximate leave-fold-out refit) and evaluating the validation loss under the updated parameters; the reported CV curve is the average of these fold estimates.

## J.4 Details for Cross-Validation

We have performed a leave-$k$-out CV to estimate the test loss. In particular, we have trained the previously mentioned two-layer model on the Friedman dataset from App. J.1.2, and set the width of the middle layer to be 100. In our experiments, we have fixed the $L_1$ regularization coefficient on $10^{-2}$, the learning rate on $10^{-4}$, the batch size on 32, and we have further trained the model with SGD with added weight decay of strength $10^{-4}$ (added in addition to the $L_1$ regularization). Our (full and approximate) CV estimates are over five different folds. For the CV approximation, we have calculated quantities of the form $u = \mathbf{F}^{-1} v$ (correspndingly for $u = \mathbf{H}^{-1} v$) by solving the linear system $\mathbf{F} u = v$ ($\mathbf{H} u = v$) and where we have used `PyTorch`'s autograd for calculating quantities of the form $\mathbf{F} u$ ($\mathbf{H} u$). The linear system was solved using `scipy.sparse.linalg.cg` running with maximum number of iterations set to 5000.

