# OpenReview forum: "Influence Estimation in Statistical Models Using the Fisher Information Matrix"
_TMLR — Accepted by TMLR_

### Review · Reviewer_uV6w · 2025-12-13

**Summary Of Contributions:**

The paper proposes the "Approximate Fisher Influence Function" (AFIF). The authors cast influence estimation as a weighted empirical risk minimization problem, deriving an estimator that replaces the inverse Hessian ,based on the infinitesimal jackknife, with the inverse Fisher Information Matrix (FIM). The paper shows that in sufficiently smooth problems, the Fisher's approximation error to the Hessian can be bounded, and presents a unified notation for defining influence estimation problems in three domains (unlearning, fairness, cross-validation). Finally, the paper shows empirically that FIM-based estimation is more stable than Hessian-based estimation in non-convex settings.

**Audience:**

Yes

**Audience Explanation:**

While most of the paper's results that would be of interest to the general machine learning community are already present in prior works, the approximation results and the extension to non-smooth regularizers might be of interest to people working on the theory of influence estimation. However, the lack of evidence of scalability seems to limit the potential interest even for this group.

**Claims And Evidence:**

No

**Claims Explanation:**

The paper's low-level are as far as I can tell correct, but the presentation of the contributions in my view exaggerates their novelty relative to what was already known, and the empirical and theoretical results do not tell a coherent story. In particular:

1. The finding that Fisher-based influence is more stable than Hessian-based influence in deep learning is presented as one of the main empirical contributions of the paper, but has already been well-established phenomenon in the literature, e.g. Bae et al. (2022) and Grosse et al. (2023), where it has been noted that non-PSD hessians can lead to unstable influence estimation.
2. The paper emphasizes the increased generality of the approach due to its application to proximal operators with non-differentiable regularizers, but does not provide evidence for the effectiveness of the approach on non-trivial problem settings, as it is tested only on a somewhat contrived synthetic setup with added noise features.
3. As far as I can tell, the "unified framework" proposed in the paper, which the authors contrast with prior "ad hoc" work, appears to be a fairly standard presentation of influence functions, and I am not sure if the paper's approach of unifying notation across domains really qualifies as a framework. As I do not have an extensive background in influence estimation it is possible that I have mischaracterized the literature on this point, however.

Finally, it seems to me that there is a fundamental disconnect between the paper's theoretical and empirical results. The theory shows that the Fisher can approximate the Hessian for parameters sufficiently close to an optimum in smooth optimization problems.  However, the empirical results say that the Fisher exhibits markedly different behaviour from the Hessian, and in particular *better* behaviour. I felt some cognitive dissonance while reading the paper to see the same estimator motivated on the one hand by being a close approximation to the Hessian in convex problems, and on the other by incurring some constant approximation error in non-convex settings that happens to bias towards more stable solutions.

**Requested Changes:**

1. Revise the presentation of the contributions to better align with the paper's actual novelty. In particular, please acknowledge that the formulation of downstream tasks via the Chain Rule on $\hat{\theta}$ is standard influence function methodology, reframe the efficiency results to make it clearer that they are summarizing existing literature and not a result of novel analysis, and explicitly state that the stability of FIM vs. Hessian is consistent with prior work (citing Bae et al., 2022 and Martens, 2020), rather than presenting it as an empirical discovery of this work.
2.  Please clarify how the optimization in the proximal operator step in Eq 9 is solved for in high-dimensional models. If the method is practically limited to small models (where the Generalized Lasso is cheap) or requires a diagonal approximation of $F$, this limitation should be stated in the abstract and introduction.
3. The use of 300 random noise features in the linear regression experiment feels contrived, and is not a convincing demonstration of the method's practicality on real data. Please either provide a stronger justification for this choice or change the experimental setup to include a more realistic source of noise in the dataset.
4. Address the conflict between the empirical and theoretical results, either by complementing the existing theoretical analysis with an analysis of settings where the FIM and Hessian disagree or by adding at least expanding on the discussion of the differences between the setting in which the theoretical results are demonstrated and the setting used for empirical evaluations.

---

> ### Author Response · Authors · 2026-01-29
>
> We thank the reviewer for the thoughtful suggestions and the detailed review.
>
> We agree that, in the current draft, the narrative connection between our experimental section and the paper’s core contribution could be clearer. To address this and to better highlight the contribution of our work, we have made the following changes:
>
> Clearer framing of the contribution. We revised the abstract, the contributions paragraph, and Section 3 to more directly reflect where our main contribution lies. In particular, our primary contribution is an accuracy guarantee for a Fisher-Information–based influence estimator in settings that may involve non-differentiable regularizers, and we now frame the results accordingly.
>
> Below we provide detailed responses to the reviewer’s additional comments:
>
> Computation / proximal operator. Our estimator requires only a single evaluation of the proximal operator. In many common cases this is computationally feasible (and standard). Moreover, similar proximal computations have been used in prior theoretical work on influence-style approximations; see, e.g., [1] and [2].
>
> Aligning experiments with the theoretical message. In line with the revised framing above, we updated the experimental section to better complement the analysis. Specifically, we added an experiment that satisfies the theoretical assumptions and demonstrates that the Fisher-based and Hessian-based estimators agree when the assumptions hold. Then, our second experiment aimed at illustrating a practical advantage of the Fisher-based approach in certain regimes—namely, that it is guaranteed to be PSD, which can yield more stable behavior than Hessian-based estimates in settings where the Hessian may be indefinite or poorly conditioned.
>
> We again thank the reviewer for taking the time to review our work. The comments were helpful in improving both the clarity and the presentation of the manuscript.
>
> [1] Wilson, Ashia, Maximilian Kasy, and Lester Mackey. "Approximate cross-validation: Guarantees for model assessment and selection." International conference on artificial intelligence and statistics. PMLR, 2020.
>
> [2] Suriyakumar, Vinith, and Ashia C. Wilson. "Algorithms that approximate data removal: New results and limitations." Advances in Neural Information Processing Systems 35 (2022): 18892-18903.

---

### Review · Reviewer_R841 · 2025-12-15

**Summary Of Contributions:**

This paper presents the Approximate Fisher Influence Function (AFIF), a new framework designed to estimate the influence of training data points within statistical models. It provides the first theoretical basis for using the Fisher Information Matrix (FIM) in place of the Hessian for influence estimation, even when dealing with non-differentiable regularizers. The approach reformulates influence estimation as a weighted empirical risk minimization problem, making it suitable for applications such as cross-validation, machine unlearning, data attribution, and fairness assessment. The study illustrates that FIM-based approximations are more computationally efficient and stable compared to Hessian-based methods, particularly in high-dimensional and non-convex contexts. Experiments demonstrate AFIF's effectiveness across various tasks, including fairness-aware unlearning and cross-validation, highlighting its advantages in terms of runtime and stability.

**Audience:**

Yes

**Audience Explanation:**

TMLR's audience would be interested in this paper, as it presents a theoretically sound, computationally efficient, and widely applicable framework for influence estimation that tackles key challenges in interpretability, fairness, and scalability.

**Claims And Evidence:**

Yes

**Claims Explanation:**

The paper's claims are supported by rigorous theory (Section 4), computational analysis (Section 3.1), and empirical validation (Section 5), providing clear and convincing evidence for the proposed method’s advantages in efficiency, stability, and applicability.

**Requested Changes:**

1. While the FIM is positive semidefinite even in non-convex settings, the theoretical analysis is primarily focused on convex problems. More discussion on non-convex guarantees would strengthen the contribution.

2. Experiments are limited to relatively small networks and datasets. Validation on larger-scale models would enhance impact.

3. Were there any cases where Hessian-based methods outperformed AFIF in terms of influence accuracy, not just runtime?

4. The theoretical results are based on several strong assumptions that might not be applicable in all real-world deep learning situations. It would be beneficial to include some remarks to address this issue.

---

> ### Author Response · Authors · 2026-01-29
>
> We thank the reviewer for the questions and comments, which we will address in order:
>
> 1 and 2. Our theoretical framework and guarantees targeted a convex setting, as done in multiple prior works in influence functions (see, for example, [1-5]). Thus, our analysis and experiments have been targeted at these settings mostly, as those lies within the framework analyzed in our work. To clarify this, we have added remark 9, which clarifies this point.
>
> 3. As discussed throughout our work, practically, the Hessian might suffer from invertibility and stability issues, which are resolved using the FIM. This is demonstrated in Fig. 2 in the revised manuscript. In the convex setting, the Hessian-based influence is known to provide strong guarantees in multiple influence measurement settings. In that regard, our work shows that influence measurement with the FIM can match the asymptotic dependence whenever $\bar{E}_n = O(1/n)$, expected to asymptotically perform similarly.
>
> 4. We agree with the determination of the reviewer regarding the validity of the assumptions made throughout our analysis in practical settings. However, assumptions of a similar nature are standard in classical influence-function analyses. To clarify both (i) how our assumptions relate to those used in prior work and (ii) where we introduce additional requirements, we added Remark 7, which explicitly compares our assumptions to those in closely related influence-function results and explains why the added assumptions are reasonable in the regimes we target.
>
> [1] Giordano, Ryan, et al. "A awiss army infinitesimal jackknife."
>
> [2] Sekhari, Ayush, et al. "Remember what you want to forget: Algorithms for machine unlearning."
>
> [3] Wilson, Ashia, et al. "Approximate cross-validation: Guarantees for model assessment and selection."
>
> [4] Stephenson, William, and Tamara Broderick. "Approximate cross-validation in high dimensions with guarantees."
>
> [5] Suriyakumar, Vinith, and Ashia C. Wilson. "Algorithms that approximate data removal: New results and limitations."

---

### Review · Reviewer_AR9z · 2026-01-16

**Summary Of Contributions:**

The work formulates a weighted ERM to define some inference objectives, namely cross
validation, fairness, machine unlearning, and data attribution.
For these, it derives some theoretical guarantees for an
approximation through influence functions
where the Hessian is approximated using
the Fisher Information Matrix,
under some smoothness and convexity assumptions.
The work shows the relation of the FIM and the Hessian from prior work,
and discusses the computational advantages of estimating the influence functions using FIM-based LiSSA.
It defines influence functions based on a proximal operator to allow for non-smooth regularizers.
Two basic experiments are conducted to show the comparable performance and
improved runtime of FIM-based LiSSA for influence functions, one for machine
unlearning and fairness, and another one for leave-one-out cross validation.

### Strengths

- The FIM is very commonly used to approximate the Hessian in Influence Functions.
  Providing theoretical guarantees for this approximation is of high interest to the community.
- The work provides a detailed appendix with extended proofs, experimental
  setups, and some additional ablation studies.
- Extending influence functions for non-differentiable regularizers is also of
  interest to the community.

### Weaknesses

- The work mostly ignores prior work on FIM/GGN-based influence functions,
  although some are briefly referenced in the introduction and conclusion.
  This is especially bad in Section 3, which introduces common practices
  as the work's own contribution.

- The experiments are fairly limited, and appear to miss some structure. Data
  attribution is missed despite being likely the most popular application of
  influence in recent years.

- The work concludes in the experiments that the FIM "achieves similar or improved
  utility over Hessian-based techniques", which some contradicts previous
  results (Hong et al., 2025).

- The work claims to clarify how FIM-based influence functions arise naturally,
  yet this observation is based on well-known results (Martens, 2020; Kunstner et al., 2019).

- The abstract only vaguely describes the contents of the paper. It does not
  state its main contribution explicitly (i.e., theoretical guarantees for
  FIM-based influence functions).

**Audience:**

Yes

**Audience Explanation:**

I think the theoretical guarantees may be of some interest to the community.
However, as the work in its current form does not sufficiently relate
the findings of prior work on FIM-based influence functions,
they may be very inaccessible.

**Broader Impact Concerns:**

I do not see any ethical implications of this work that would require a broader impact statement.

**Claims And Evidence:**

No

**Claims Explanation:**

The work is not clearly stating which parts are its own contributions,
and which parts are prior work.
Especially Section 3 appears to claim FIM-based LiSSA for influence functions
as the work's own contribution, despite its use in prior work (Bae et al.,
2022).
The experimental results on the fairness/unlearning experiment are hard to
interpret as evidential for the claim that FIM-based influence provides a
"similar or improved utility" compared to Hessian-based techniques.

I did not carefully check the proofs.

**Requested Changes:**

1. (critical) Section 3 is very vague about the contributions of this work. The FIM is equivalent to the GGN (Martens et al., 2014) for loss functions that are the log-likelihood of exponential families, as studied in this work. As stated in the introduction, multiple works have leveraged the FIM/GGN for influence functions, including Barshan et al. (2020) and Teso et al. (2021) as the FIM, and Bae et al. (2022) as the GGN. Barshan et al. (2020) also motivate IF with weighted ERM. However, the work quickly discards these relations by claiming they only consider the "empirical FIM". But this is problematic, as both Barshan et al. (2020) and Teso et al. (2021) formulate the FIM similarly as an expectation over the model's predictive distribution (it is more explicit in Teso et al. (2021) in (13)), which is the "approximate Fisher" discussed in this work. It is critical for this work to discuss all prior work on FIM/GGN-based influence functions in detail and make clear where its contributions lie.

2. (critical) The definition of the "empirical Fisher" and "approximate Fisher" seems problematic. The empirical Fisher as introduced by Kunstner et al. (2019) (see Equation 6) in statistics is in fact what is referred to as the "approximate Fisher" in this work. The definition needs to be made clear, and it should be clarified which prior work uses which.

2. (critical) Remark 4 appears to assume that LiSSA with GGN/FIM has not been done without computing the full FIM before, but this is not correct. For instance, Bae et al. (2022) implement LiSSA with the GGN as in (7). Although the manuscript itself lacks some detail in this regard, the implementation clearly shows this. It is critical to clarify the work's contribution here.
  - see [here](https://github.com/alstonlo/torch-influence/blob/8b4f0756acf642a38a8b24a517da89343a9de7d0/torch_influence/modules.py#L253) and [here](https://github.com/alstonlo/torch-influence/blob/8b4f0756acf642a38a8b24a517da89343a9de7d0/torch_influence/base.py#L376), which implements both (7) and (8)

3. (critical) The work claims to introduce (in the abstract, and Section 3) how FIM-based influence functions arise naturally. However, it appears that the only evidence for this claim seems to be that the FIM approximates the Hessian due to the decomposition of the Hessian as shown by Schraudolph et al. (2002) and others, and the observation the term added to the FIM shrinks as training accuracy improves (Martens, 2020; Kunstner et al., 2019). This is a well known result, and discussed in the context of influence functions, for example, by Barshan et al. (2020) (in Appendix A). This needs to be discussed in more detail, or removed from the contributions.

4. (critical) While the proximal formulation in Sec. 4.2 does remind me of Bae et al. (2022), I am not aware of any work about non-differentiable regularizers in the context of influence functions, and think that is an important point of the work that should be emphasized. Unfortunately, this is discussed only in a half-sentence as arising due to the formulation of Influence Functions from the proximal operator, despite being introduced as a main contribution. I do not think the brief motivation in remark 5 and the empirical demonstration of the L1 regularizer in the short experimental section are enough. The work needs to address this in more detail.

5. (critical) The abstract only vaguely describes the contents of the paper. It does not state explicitly that it provides theoretical guarantees influence functions using a FIM-based approximation of the Hessian, despite that being its main contribution.

6. (critical) The experiments are fairly limited and lack some structure.
  - Figure 1 is hard to interpret, especially as in the Adult setting, neither method appears very performant. This makes it very difficult to use as evidence for that claim that the FIM-based influence performs at least as good as the Hessian. For this, it would help considerably to do some pair-wise significance test between Hessian and Fisher.
  - It would help the work greatly to split the experiments into the three claims made in the introduction of the experiments. First, test the performance to verify that the FIM can in fact replace the Hessian safely. Second, show that using the FIM reduces computation time. Third, show an experiment where both a differentiable, and a non-differentiable regularizer are used, ideally in a setting where the non-differentiable regularizer leads to a more performant model (as in 5.2). If performance and runtime could be shown with the same metrics on different applications (fairness, unlearning, data attribution, cross validation, ...) on multiple datasets, it would make a strong point.
  - The gain over using the FIM might show more in more complex models than the ones analyzed, where the Hessian starts to become degenerate and the FIM remains PSD.

7. (critical) The work comes to the conclusion that the FIM "achieves similar or improved utility as the Hessian-based techniques" for influence functions. This contradicts the results in Hong et al. (2025), who directly compare different approximations for the Hessian in influence functions, including the GGN, coming to the conclusion that the Hessian provides the most utility. The reason for this contradiction is not clear. For instance, it could stem from the approximation of the empirical FIM in that work not averaging over the model's conditional predictive distribution. A similar empirical ablation would help the point of this work greatly.

8. (stronger) Data attribution is currently likely the most prominent application of Influence functions, yet the empirical experiments skip it. This limits its comparability with other works such as Hong et al. (2025), who conduct a direct comparison of Hessian approximations for data attribution.

9. (stronger) In "Comparing Computations" in 3.3.1, it is stated "evaluating (7) requires just one differentiation in backward mode", which can be a misleading statement. Below (7), it is stated that one VJP and on JVP is required, where the VJP (one backward diff) needs to be evaluated first, followed by the HVP of $\nabla_f y$, followed by the JVP (one forward diff). While cheaper in memory, and slightly cheaper in computation, VJP will still require an additional forward, after the backward and forward of the VJP. The work should clarify this.

10. (stronger) In 2.1.1, the work introduces two ways to approximate the influence to an inference objective, one via directly substituting the approximation $\tilde\theta(w^n)$ into the inference objective, and one by linearising the inference objective, in which the same approximation of the parameters arises. It seems that the work mostly focuses on these plug-in estimates. But in practice, using the plug-in estimator on non-linear inference objectives, like the loss of neural networks, may yield very unexpected results when the magnitude of $\tilde\theta(w^n)$ is not on-point, e.g., when the approximation of the Hessian has some error. It would help to clarify why the plug-in estimator is preferred over the linearised one.

**References:**

- Elnaz Barshan, Marc-Etienne Brunet, and Gintare Karolina Dziugaite. 2020. “RelatIF: Identifying Explanatory Training Samples via Relative Influence.” Proceedings of the Twenty Third International Conference on Artificial Intelligence and Statistics, June 3, 1899–909. https://proceedings.mlr.press/v108/barshan20a.html.
- Stefano Teso, Andrea Bontempelli, Fausto Giunchiglia, and Andrea Passerini. 2021. “Interactive Label Cleaning with Example-Based Explanations.” Advances in Neural Information Processing Systems 34: 12966–77. https://proceedings.neurips.cc/paper/2021/hash/6c349155b122aa8ad5c877007e05f24f-Abstract.html.
- Steve Hong, Runa Eschenhagen, Bruno Mlodozeniec, and Richard Turner. 2025. “Better Hessians Matter: Studying the Impact of Curvature Approximations in Influence Functions.” arXiv:2509.23437. Preprint, arXiv, September 27. https://doi.org/10.48550/arXiv.2509.23437.

---

> ### Author Response · Authors · 2026-01-29
>
> We thank the reviewer for the thorough and insightful review. We have made our best effort to address each point and to revise the manuscript accordingly in response to the concerns raised.
>
> Comments regarding critical points:
>
> 1. We have modified section 3 and its structure. In particular, we have framed it as a survey of previous results, which aim to provide background and motivation for our new accuracy guarantees, which are derived in SectionW 4.
>
> 2. We thanks the reviewer for pointing out our concerns regarding the definitions made in Kunstner et al. (2019). However, we aimed to follow these definitions in our work. In particular, Eq (7) in Kunstner et al. (2019) refers to the "empirical Fisher" as the FIM where we average over both $\left\{y_n\right\}$ and $\left\{x_n\right\}$, and where the "Fisher" matrix is the one obtained by averaging over the $\{x_n\}$ and taking expectation over $P_{y|x;\theta}$, and which matches the definitions are made in our work (see eq (5)). To further clarify our definitions, we have added remark 3, which clarifies the different definitions.
>
> 3. We have clarified remark 4 to avoid confusion regarding the contributions made by our work. In particular, we agree and acknowledge that the FIM has been used in previous works (for example, in [1]). However, our contribution lies in the new accuracy guarantees and the extension to settings with non-differentiable regularization.
>
> 4. We have added more content regarding the importance of the inclusion of a general regularization in our formulation. In particular, we have: 1. added additional discussion in the introduction regarding the importance of non-differentiable regularization in multiple problems in machine learning, 2. expanded on remark 8, claiming the practicality of the calculation of the proximal operator and the connection to previous works, and 3. added an additional experiment with an $\ell_1$ regularizer. We believe that this extends the importance and weight of this discussion regarding the inclusion of non-differentiable regularizations. We thank the reviewer for mentioning this and suggesting this improvement!
>
> 5. We have changed the abstract to reflect the new structure and framework of the paper, which aims to clarify the key contributions we provide. In particular, we emphasize that our main contribution is new accuracy guarantees that extend to cases with non-differentiable regularizers. We have further made similar changes throughout the work, to help clarify where our work reside related to other works in the literature on influence functions.
>
> 6. Based on the comments given and the new framework of the paper, we have changed the experiments to reflect more on the contributions we have made. In particular, we have switched to having two experiments, where one experiment falls within the theoretical framework we analyze in our work and is intended to demonstrate the validity of our theoretical analysis, showing the the FIM can be used to replace the Hessian but is usually more comutationally efficient and a second experiment aim to demonstrate the application of the FIM in a more general setting that slightly deviate from the theoretical assumptions. Both settings include a non-differentiable regularization, which is justified by generating target solutions that are sparse. We believe that this new setting communicates the contributions made by our work in a better way. We thank the reviewer for suggesting the modified set of experiments.
>
> 7. As mentioned throughout, our main contribution relies on the new guarantees derived, which are applicable in settings with non-differentiable regularization. In that regard, the theory predicts that under some technical assumptions these guaratnees matches those exists for the Hessian. This is different than the setting studies by Hong et al. (2025), which empirically study a non-convex setting. Additionally, as mentioned in Hong et al. (2025), the Hessian might be ill-conditioned, which is tyically address by adding some form of regularization. However, this provides an additional hyperparameter that the designer needs to set, which in certain cases might lead to drastically different results. Furthermore, Hong et al. (2025) do not concern computational efficiency. Thus, the main claim (which is supported by our proofs) we make is regarding the fact that the FIM can replace the Hessian under the theoretical framework considered in our work, and where further in practical settings it has the benefit of being PSD and more computationally efficient, thus in certain cases might lead to improved estimations in these cases as well. We thank the reviewer for making this clarification!
>
> [1] Bae, Juhan, et al. "If influence functions are the answer, then what is the question?." Advances in Neural Information Processing Systems 35 (2022): 17953-17967.

---

> ### Author Response · Authors · 2026-01-30
>
> Comments regarding stronger points:
>
> 8. We agree that data attribution is an important and active area in the influence estimation space. Our focus, however, is on establishing accuracy guarantees for the proposed estimator, and we have restructured the experiments to better reflect and test the specific claims of our analysis. With respect to Hong et al. (2025), we want to clarify that their scope is different: they investigate FIM-based influence estimators primarily through empirical evaluation in a different problem formulation, whereas our work targets a separate theoretical setting and provides accuracy guarantees (including cases involving non-differentiable regularizers). Because the objectives and assumptions do not align cleanly, a direct comparison would be potentially ambiguous; We see our theoretical analysis as complementary to their work.
>
> 9. Thank you for raising this subtlety — we agree that the phrasing “just one differentiation in backward mode” can be misread as a full accounting of all AD passes needed to evaluate (7). We have therefore revised the “Comparing Computations” paragraph to make the claims explicit. In particular, a VJP in standard reverse-mode AD consists of one forward evaluation (to build/record the computation) followed by one backward sweep; importantly, it does not inherently require an additional forward pass after the backward. This is also reflected in the PyTorch reference implementation of "torch.autograd.functional.vjp" in [2], where the function is evaluated exactly once via "outputs = func(*inputs)", and the VJP is then obtained via a single reverse-mode "call grad_res = _autograd_grad(outputs, inputs, v, ...)" (see [2]). An extra forward can arise in practice only when one invokes VJP/JVP routines in a way that recomputes "func(*inputs)" (for example, for subsequent gradient computations). We have added a clarification regarding the computations in the text. We appreciate the reviewer highlighting this point and have updated the exposition accordingly (see [1,2]).
>
> 10. Thank you for raising this point. We agree that for highly non-linear inference objectives, a plug-in evaluation $g(\widetilde{\theta})$ can in principle behave poorly if $\widetilde{\theta}$ is not sufficiently accurate (e.g., due to error in the Hessian/FIM approximation). In that context, we want to first highlight that our analysis (in particular Theorem. 1) is stated for both options of the approximation, and thus the theoretical results are stated in terms of both. Second, we use the plug-in estimator in our experiments because it avoids introducing an additional Taylor (truncation) approximation of the inference objective $g$. The linearized alternative can be useful when $\||\widetilde{\theta}-\widehat{\theta}\||$ is provably small and $g$ is well-approximated locally, but it can also be inaccurate when higher-order terms are non-negligible. We note that this is also the setting in [3], which approximates CV using plug-in estimates.
>
> [1] https://iclr-blogposts.github.io/2024/blog/bench-hvp/
>
> [2] https://github.com/pytorch/pytorch/blob/v2.10.0/torch/autograd/functional.py#L270
>
> [3] Wilson, Ashia, et al. "Approximate cross-validation: Guarantees for model assessment and selection."

---

### Author Response · Authors · 2026-01-29

We'd like to thank all reviewers for their constructive feedback and questions. We’ve carefully addressed each point in our responses and have revised the paper draft accordingly. All updated sections are highlighted in blue for clarity.

We look forward to continued discussion and are happy to address any further questions or suggestions.

---

### Decision · Action_Editor_mpac · 2026-03-19

**Recommendation:** Accept as is

**Audience:**

Yes

**Audience Explanation:**

Influence function is an important topic in machine learning and deep learning.

**Claims And Evidence:**

Yes

**Claims Explanation:**

Many concerns were raised by the reviewers but the rebuttal has answered them satisfactorily. They all agree that the paper has improved generally. One reviewer believes that the paper "requires some more polish, as it reads like patch work in places (for instance, contributions are repeated)". The authors can easily address this to further improve the quality of their work.